# ON GENERALIZATION BOUNDS OF A FAMILY OF RECURRENT NEURAL NETWORKS

## ABSTRACT

Recurrent Neural Networks (RNNs) have been widely applied to sequential data analysis. Due to their complicated modeling structures, however, the theory behind is still largely missing. To connect theory and practice, we study the generalization properties of vanilla RNNs as well as their variants, including Minimal Gated Unit (MGU) and Long Short Term Memory (LSTM) RNNs. Specifically, our theory is established under the PAC-Learning framework. The generalization bound is presented in terms of the spectral norms of the weight matrices and the total number of parameters. We also establish refined generalization bounds with additional norm assumptions, and draw a comparison among these bounds. We remark: (1) Our generalization bound for vanilla RNNs is significantly tighter than the best of existing results; (2) We are not aware of any other generalization bounds for MGU and LSTM in the exiting literature; (3) We demonstrate the advantages of these variants in generalization.

## 1 INTRODUCTION

Recurrent Neural Networks (RNNs) have successfully revolutionized sequential data analysis, and been widely applied to many real world problems, such as natural language processing (Cho et al., 2014; Bahdanau et al., 2014; Sutskever et al., 2014), speech recognition (Graves et al., 2006; Mikolov et al., 2010; Graves, 2012; Graves et al., 2013), computer vision (Gregor et al., 2015; Xu et al., 2015; Donahue et al., 2015; Karpathy and Fei-Fei, 2015), healthcare (Lipton et al., 2015; Choi et al., 2016a;b), and robot control (Ku and Lee, 1995; Lee and Teng, 2000; Yoo et al., 2006). Quite a few of these applications can be approached easily in our daily life, such as Google Translate, Google Now, Apple Siri, etc.

The sequential modeling nature of RNNs is significantly different from feedforward neural networks, though they both have neurons as the basic components. RNNs exploit the internal state (also known as hidden unit) to process the sequence of inputs, which naturally captures the dependence of the sequence. RNNs can also be viewed as nonlinear dynamical systems, and reduced to linear dynamical systems given identity activation operators. Besides the vanilla version, RNNs have many other variants. A large class of variants incorporate the so-called "gated" units to trim RNNs for different tasks. Typical examples include Long Short-Term Memory (LSTM, Hochreiter and Schmidhuber (1997)), Gated Recurrent Unit (GRU, Jozefowicz et al. (2015)) and Minimal Gated Unit (MGU, Zhou et al. (2016)).

The success of RNNs owes not only to their special network structures and the ability to fit training data, but also to their good generalization property: They can provide accurate predictions on unseen data. For example, Graves et al. (2013) report that after training with merely 462 speech samples, deep LSTM RNNs achieve a test set error of 17.7% on TIMIT phoneme recognition benchmark, which is the best recorded score. Mikolov et al. (2010) also show that RNNs outperform significantly state-of-the-art backoff models for speech recognition. When using RNNs in Wall Street Journal task to predict the next word in textual data given the context, word error rate is reduced around 18% compared to backoff models trained on the same amount of data, and 12% when backoff model is trained on 5 times more data. Despite of the popularity of RNNs in applications, their theory is less studied than other feedforward neural networks (Bartlett et al., 2017; Neyshabur et al., 2017; Golowich et al., 2017; Li et al., 2018). There are still several long lasting fundamental questions regarding the approximation, trainability, and generalization of RNNs.

In this paper, we propose to understand the generalization ability of RNNs and their variants. We aim to answer two questions from a theoretical perspective:

*Q.1) Do RNNs suffer from significant curse of dimensionality?*

*Q.2) What are the advantages of MGU and LSTM over vanilla RNNs?*

The investigation of generalization properties of neural networks including RNNs has a long history. Most of these works adopt a layer wise analysis and establish their results by induction. For example, Haussler (1992) establishes a complexity bound for feedforward neural networks. The result assumes average Lipschitz constant of each layer to be greater than 1 and involves the product of these Lipschitz constants. Thus, the resulting bound is inevitably exponential in the depth of the network. Later, Dasgupta and Sontag (1996) and Koiran (1998) adopt a VC-dimension argument to show complexity bounds of RNNs that are polynomial in the size of the network. Both of the works, however, are based on oversimplified assumptions: Dasgupta and Sontag (1996) only consider linear RNNs for binary classification tasks; Koiran (1998) assumes RNNs take the first coordinate of their hidden states as outputs. More recently, Bartlett et al. (2017) propose a new technique for developing generalization bounds for feedforward neural networks based on empirical Rademacher complexity under the PAC-Learning framework. Neyshabur et al. (2017) further adapt the technique to establish their generalization bound using the PAC-Bayes approach. Then the follow-up work Zhang et al. (2018) use the PAC-Bayes approach to establish a generalization bound for vanilla RNNs.

Our theory is partially motivated by Bartlett et al. (2017), but is quite different from Bartlett et al. (2017) and Zhang et al. (2018). In particular, our analysis exploits the compositional nature of RNNs, and decouples the spectral norms of weight matrices and the number of weight parameters. This makes our analysis conceptually much simpler (e.g. avoid layer wise analysis), and also yields better generalization bound than Zhang et al. (2018).

Taking a sequence to sequence multiclass classification problem as an example, we observe $m$ sequences of data points $(x_{i,t}, z_{i,t})_{t=1}^T$, where $x_{i,t} \in \mathbb{R}^{d_x}$ and the class label $z_{i,t} \in \{1, \ldots, K\}$ for all $t = 1, ..., T$ and $i = 1, ..., m$. Each sequence is drawn independently from some underlying distribution over $\mathbb{R}^{d_x \times T} \times \{1, \ldots, K\}$. The vanilla RNNs compute $h_{i,t}$ and $y_{i,t}$ iteratively as follows,

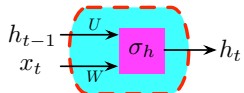

Figure 1: A basic building block of vanilla RNNs

$$h_{i,t} = \sigma_h \left( U h_{i,t-1} + W x_{i,t} \right), \quad \text{and} \quad y_{i,t} = \sigma_y \left( V h_{i,t} \right),$$

where $\sigma_y$ and $\sigma_h$ are activation operators, $h_{i,t} \in \mathbb{R}^{d_h}$ is the hidden state with $h_{i,0} = 0$, and $U \in \mathbb{R}^{d_h \times d_h}$, $V \in \mathbb{R}^{d_y \times d_h}$, and $W \in \mathbb{R}^{d_h \times d_x}$ are weight matrices. The activation operators $\sigma_h$ and $\sigma_y$ are entrywise, i.e., $\sigma_h([v_1, \ldots, v_d]^\top) = [\sigma_h(v_1), \ldots, \sigma_h(v_d)]^\top$, and Lipschitz with parameters $\rho_h$ and $\rho_y$ respectively. For simplicity, we assume $\sigma_h(\cdot) = \tanh(\cdot)$, $\sigma_y(0) = 0$, and $\rho_y = 1$. Extensions to general activation operators are given in Section 2. For a new testing sequence $(x_t, z_t)_{t=1}^T$, we predict the label sequence using

$$\widetilde{z}_t = \mathrm{argmax}_j [y_t]_j, \quad \text{for all } t = 1, \ldots, T.$$

To establish the generalization bound, we need to define the "model complexity" of vanilla RNNs. In this paper, we adopt the empirical Rademacher complexity (ERC, see more details in Section 2), which has been widely used in the existing literature on PAC-Learning. For many nonparametric function classes, we often need complicated argument to upper bound their ERC. Our analysis, however, shows that we can upper bound the ERC of vanilla RNNs in a very simple manner by exploiting their Lipschitz continuity with respect to (w.r.t) the model parameters, since they are essentially in parametric forms. More specifically, denote $\mathcal{F}_t = \{f_t : \{x_1, ..., x_t\} \mapsto y_t\}$ as the class of mappings from the first $t$ inputs to the $t$-th output computed by vanilla RNNs. For a matrix $A$, $\|A\|_2$ denotes the spectral norm, and for a vector $v$, $\|v\|_2$ denotes the Euclidean norm. Define $\frac{x^t - 1}{x - 1} = t$ for $x = 1$. Then, informally speaking, the "model complexity" of vanilla RNNs satisfies

$$\mathsf{Complexity} = O \left( d \min \left\{ \sqrt{d}, \|W\|_2 \frac{\|U\|_2^t - 1}{\|U\|_2 - 1} \right\} \|V\|_2 \sqrt{\log \left( t \sqrt{d} \frac{\|U\|_2^t - 1}{\|U\|_2 - 1} \right)} \right),$$

where $d = \sqrt{d_x d_h + d_h^2 + d_h d_y}$. We then give the generalization bound in the following statement.

**Theorem 1** (informal). Given a collection of samples $S = \big\{(x_{i,t}, z_{i,t})_{t=1}^T, i = 1, ..., m\big\}$ with $\|x_{i,t}\|_2 \leq 1$ and a new testing sequence $(x_t, z_t)_{t=1}^T$, with probability at least $1 - \delta$ over $S$, for every margin value $\gamma > 0$ and $f_t \in \mathcal{F}$ for integer $t \leq T$, we have,

$$\mathbb{P}(\widetilde{z}_t \neq z_t) \leq \widehat{\mathcal{R}}_{\gamma,t} + O\left(\frac{\mathsf{Complexity}}{\sqrt{m}\gamma} + \sqrt{\frac{\log \frac{1}{\delta}}{m}}\right), \tag{1}$$

where $\widehat{\mathcal{R}}_{\gamma,t} = \frac{1}{m} \sum_{i=1}^m \mathbb{1}([y_{i,t}]_{z_{i,t}} \leq \max_{j \neq z_{i,t}} [y_{i,t}]_j + \gamma)$.

Please refer to Section 2 for a complete statement. The generalization bound in Theorem 1 can be interpreted under three different scenarios[1]: **(I)** When $\|U\|_2 < 1$, the generalization bound is $\widetilde{O}\big(\frac{d}{\sqrt{m}\gamma}\big)$, which only has a logarithmic dependence on $t$; **(II)** When $\|U\|_2 = 1$, the generalization bound is $\widetilde{O}\big(\frac{dt}{\sqrt{m}\gamma}\big)$, which has a linear dependence on $d$ and $t$; **(III)** When $\|U\|_2 > 1$, the generalization bound is $\widetilde{O}\big(\frac{\sqrt{d^3t}}{\sqrt{m}\gamma}\big)$, which has a polynomial dependence on $d$ and $t$.

We theoretically justify that vanilla RNNs do not suffer from significant curse of dimensionality. Because they compute outputs $y_t$ recursively using the same weight matrices, and their hidden states $h_t$ are entrywise bounded.

Compared with the generalization bound in Zhang et al. (2018), which is of the order

$$\widetilde{O}\left(\frac{dt^2 \|W\|_2 \|V\|_2 \max\{1, \|U\|_2^t\}}{\sqrt{m}\gamma}\right),$$

our bound is tighter by a factor of $t^2$ for case **(I)**, a factor of $t$ for case **(II)**. Additionally, Zhang et al. (2018) fail to incorporate the boundedness condition of hidden state into their analysis, thus the generalization bound is exponential in $t$ for case **(III)**. Our generalization bound, however, is still polynomial in $d$ and $t$ for case **(III)**.

Moreover, **(II)** is closely related to a few recent results on imposing orthogonal constraints on weight matrices to stabilize the training of RNNs (Saxe et al., 2013; Le et al., 2015; Arjovsky et al., 2016; Vorontsov et al., 2017; Zhang et al., 2018). We remark that from a learning theory perspective, **(II)** also implies that the orthogonal constraints can potentially help generalization.

We also present refined generalization bounds with additional matrix norm assumptions. These assumptions allow us to derive norm-based generalization bounds. We draw a comparison among these bounds and highlight their advantage under different scenarios.

Our theory can be further extended to several variants, including MGU and LSTM RNNs. Specifically, we show that the gated units in MGU and LSTM RNNs can introduce extra decaying factors to further reduce the dependence on $d$ and $t$ in generalization. Such an advantage in generalization make these RNNs do not suffer from significant curse of dimensionality either. To the best of our knowledge, these are the first results on generalization guarantees for these neural networks.

The rest of the paper is organized as follows: Section 2 presents the generalization bound of vanilla RNNs; Section 3 presents the proof outline of the generalization bound; Section 4 presents refined generalization bounds and their comparison; Section 5 presents the generalization bound of MGU and LSTM RNNs; Section 7 discusses related works and collects open problems.

**Notations**: Given a vector $v \in \mathbb{R}^d$, we denote the vector Euclidean norm by $\|v\|_2^2 = \sum_{i=1}^d |v_i|^2$, and the infinity norm by $\|v\|_\infty = \max_j |v_j|$. Given a matrix $M \in \mathbb{R}^{m \times n}$, we denote the spectral norm by $\|M\|_2$ as the largest singular value of $M$, the Frobenius norm by $\|M\|_{\mathrm{F}}^2 = \mathrm{trace}(MM^\top)$, and the $(2,1)$ norm by $\|M\|_{2,1} = \sum_{i=1}^n \|M_{:,i}\|_2$. Given a function $f$, we denote the function infinity norm by $\|f\|_\infty = \sup |f|$. We adopt the standard $O(\cdot)$ notation, which is defined as $f(x) = O(g(x))$ for $x \to \infty$ if and only if there exists $M > 0$ and $x_0$, such that $|f(x)| \leq Mg(x)$ for $x \geq x_0$. We use $\widetilde{O}(\cdot)$ to denote $O(\cdot)$ with hidden log factors.

---

[1]To ease the discussion, we assume $\|U\|_2$ does not scale with $t$. Therefore, $\|U\|_2 < 1$ is equivalent to $\|U\|_2 \leq 1 - \Delta$ for a constant $\Delta > 0$. A precise statement can be found following Theorem 2.

## 2 GENERALIZATION OF VANILLA RNNS

To establish the generalization bound, we start with imposing some mild assumptions.

**Assumption 1.** Input data are bounded, i.e., $\|x_{i,t}\|_2 \leq B_x$ for all $i = 1, \ldots, m$ and $t = 1, \ldots, T$.

**Assumption 2.** The spectral norms of weight matrices are bounded respectively, i.e., $\|U\|_2 \leq B_U$, $\|V\|_2 \leq B_V$, and $\|W\|_2 \leq B_W$.

**Assumption 3.** Activation operators $\sigma_h$ and $\sigma_y$ are Lipschitz with parameters $\rho_h$ and $\rho_y$ respectively, and $\sigma_h(0) = \sigma_y(0) = 0$. Additionally, $\sigma_h$ is entrywise bounded by $b$.

Assumptions 1 and 2 are moderate assumptions. Moreover, Assumption 3 holds for most commonly used activation operators, such as $\sigma_h(\cdot) = \tanh(\cdot)$ and $\sigma_y(\cdot) = \text{ReLU}(\cdot) = \max\{\cdot, 0\}$ (1-Lipschitz).

Recall that vanilla RNNs compute $h_{i,t}$ and $y_{i,t}$ as follows,

$$h_{i,t} = \sigma_h\left(U h_{i,t-1} + W x_{i,t}\right) \quad \text{and} \quad y_{i,t} = \sigma_y\left(V h_{i,t}\right),$$

where $U \in \mathbb{R}^{d_h \times d_h}$, $V \in \mathbb{R}^{d_y \times d_h}$, and $W \in \mathbb{R}^{d_h \times d_x}$. Given a sequence $(x_t, z_t)_{t=1}^T$, we define $X_t \in \mathbb{R}^{d_x \times t}$ by concatenating $x_1, \ldots, x_t$ as columns of $X_t$. Recall that we denote $\mathcal{F}_t = \{f_t : X_t \mapsto y_t\}$ as the class of mappings from the first $t$ inputs to the $t$-th output computed by vanilla RNNs. Then we define the functional margin for the $t$-th output in vanilla RNNs as

$$\mathcal{M}(f_t(X_t), z_t) = [f_t(X_t)]_{z_t} - \max_{j \neq z_t}[f_t(X_t)]_j.$$

We further define a ramp loss $\ell_\gamma\left(-\mathcal{M}(f_t(X_t), z_t)\right) : \mathbb{R} \mapsto \mathbb{R}^+$ to each margin, where $\ell_\gamma$ is a piecewise linear function defined as

$$\ell_\gamma(a) = \mathbb{1}\{a > 0\} + (1 + \frac{a}{\gamma})\mathbb{1}\{-\gamma \leq a \leq 0\},$$

where $\mathbb{1}\{A\}$ denotes the indicator function of a set $A$. Accordingly, the ramp risk is defined as $\mathcal{R}_\gamma(f_t) = \mathbb{E}\left[\ell_\gamma\left(-\mathcal{M}(f_t(X), z_t)\right)\right]$, and its empirical counterpart is defined as $\widehat{\mathcal{R}}_\gamma(f_t) = \frac{1}{m}\sum_{i=1}^m \ell_\gamma\left(-\mathcal{M}(f_t(X_{i,t}), z_{i,t})\right)$. We then present the formal statement of Theorem 1.

**Theorem 2.** Let activation operators $\sigma_h$ and $\sigma_y$ be given, and Assumptions 1–3 hold. Then for $(x_t, z_t)_{t=1}^T$ and $S = \left\{(x_{i,t}, z_{i,t})_{t=1}^T, i = 1, \ldots, m\right\}$ drawn i.i.d. from any underlying distribution over $\mathbb{R}^{d_x \times T} \times \{1, \ldots, K\}$, with probability at least $1 - \delta$ over $S$, for every margin value $\gamma > 0$ and every $f_t \in \mathcal{F}_t$ for integer $t \leq T$, we have

$$\mathbb{P}\left(\widetilde{z}_t \neq z_t\right) \leq \widehat{\mathcal{R}}_\gamma(f_t) + O\left(\frac{d\rho_y B_V \min\left\{b\sqrt{d}, \rho_h B_x B_W \frac{\beta^t - 1}{\beta - 1}\right\}\sqrt{\log\left(t\sqrt{dm}\frac{\beta^t - 1}{\beta - 1}\right)}}{\sqrt{m}\gamma} + \sqrt{\frac{\log\frac{1}{\delta}}{m}}\right),$$

where $d = \sqrt{d_x d_h + d_h^2 + d_h d_y}$ and $\beta = \rho_h B_U$.

We remark that the generalization bound depends on the total number of weights, and the range of $\rho_h B_U$ in three cases as indicated in Section 1. More precisely, if $\rho_h B_U \lesssim \left(1 + \frac{1}{t^\alpha}\right)$ for constant $\alpha > 0$ bounded away from zero, the generalization bound is of the order $\widetilde{O}\left(\frac{dt^\alpha}{\sqrt{m}\gamma}\right)$, which has a polynomial dependence on $d$ and $t$. As can be seen, with proper normalization on model parameters, vanilla RNNs do not suffer from significant curse of dimensionality.

We also highlight a tradeoff between generalization and representation of vanilla RNNs. As can be seen, when $\rho_h B_U$ is strictly smaller than 1, the generalization bound is nearly independent on $t$. The hidden state, however, only has limited representation ability, since its magnitude diminishes as $t$ grows large. On the contrary, when $\rho_h B_U$ is strictly greater than 1, the representation ability of hidden state is amplified but the generalization becomes worse. As a consequence, recent empirical results show that imposing extra constraints or regularization, such as $U^\top U = I$ or $\|U\|_2 \leq 1$ (Saxe et al., 2013; Le et al., 2015; Arjovsky et al., 2016; Vorontsov et al., 2017; Zhang et al., 2018), helps balance the generalization and representation of RNNs.

## 3 PROOF OF MAIN RESULTS

Our analysis is based on the PAC-learning framework. Due to space limit, we only present an outline of our proof. More technical details are deferred to Appendix A. Before we proceed, we first define the empirical Rademacher complexity as follows.

**Definition 1** (Empirical Rademacher Complexity). Let $\mathcal{H}$ be a function class and $S = \{s_1, \ldots, s_m\}$ be a collection of samples. The empirical Rademacher complexity of $\mathcal{H}$ given $S$ is defined as

$$\text{Empirical Rademacher Complexity: } \mathfrak{R}_S(\mathcal{H}) = \mathbb{E}_\epsilon \left[ \sup_{h \in \mathcal{H}} \frac{1}{m} \sum_{i=1}^m \epsilon_i h(s_i) \right],$$

where $\epsilon_i$'s are i.i.d. Rademacher random variables, i.e., $\mathbb{P}(\epsilon_i = 1) = \mathbb{P}(\epsilon_i = -1) = 0.5$.

We then proceed with our analysis. Recall that Mohri et al. (2012) give an empirical Rademacher complexity (ERC)-based generalization bound, which is restated in the following lemma with $\mathcal{F}_{\gamma,t} = \{(X_t, z_t) \mapsto \ell_\gamma(-\mathcal{M}(f_t(X_t), z_t)) : f_t \in \mathcal{F}_t\}$.

**Lemma 1.** Given a testing sequence $(x_t, z_t)_{t=1}^T$, with probability at least $1 - \delta$ over samples $S = \{(x_{i,t}, z_{i,t})_{t=1}^T, i = 1, \ldots, m\}$, for every margin value $\gamma > 0$ and any $f_t \in \mathcal{F}_t$, we have

$$\mathbb{P}(\widetilde{z}_t \neq z_t) \leq \mathcal{R}_\gamma(f_t) \leq \widehat{\mathcal{R}}_\gamma(f_t) + 2\mathfrak{R}_S(\mathcal{F}_{\gamma,t}) + 3\sqrt{\frac{\log \frac{2}{\delta}}{2m}}. \tag{2}$$

Note that Lemma 1 adapts the original version (Theorem 3.1, Chapter 3.1, Mohri et al. (2012)) for the multiclass ramp loss, and we have $\mathbb{P}(\widetilde{z}_t \neq z_t) \leq \mathcal{R}_\gamma(f_t)$ by definition.

Now we only need to bound the ERC $\mathfrak{R}_S(\mathcal{F}_{\gamma,t})$. Our analysis consists of three steps. First, we characterize the Lipschitz continuity of vanilla RNNs w.r.t model parameters. Next, we bound the covering number of function class $\mathcal{F}_t$. At last, we derive an upper bound on $\mathfrak{R}_S(\mathcal{F}_{\gamma,t})$ via the standard machinery in the PAC-learning framework. Specifically, consider two different sets of weight matrices $(U, V, W)$ and $(U', V', W')$. Given the same activation operators and input data, denote the $t$-th output as $y_t$ and $y'_t$ respectively. We characterize the Lipschitz property of $\|y_t\|_2$ w.r.t model parameters in the following lemma.

**Lemma 2.** Under Assumptions 1–3, given input $(x_t)_{t=1}^T$ and for any integer $t \leq T$, $\|y_t\|_2$ is Lipschitz in $U$, $V$ and $W$, i.e.,

$$\|y_t - y'_t\|_2 \leq L_{U,t} \|U - U'\|_{\mathrm{F}} + L_{V,t} \|V - V'\|_{\mathrm{F}} + L_{W,t} \|W - W'\|_{\mathrm{F}},$$

where $L_{U,t} = \rho_h B_V B_W t a_t$, $L_{V,t} = B_W a_t$, and $L_{W,t} = B_V a_t$ with $a_t = \rho_y \rho_h B_x \frac{(\rho_h B_U)^t - 1}{\rho_h B_U - 1}$.

The detailed proof is provided in Appendix A.2. We give a simple example to illustrate the proof technique. Specifically, we consider a single layer network that outputs $y = \sigma(Wx)$, where $x$ is the input, $\sigma$ is an activation operator with Lipschitz parameter $\rho$, and $W$ is a weight matrix. Such a network is Lipschitz in both $x$ and $W$ as follows. Given weight matrices $W$ and $W'$, we have

$$\|y - y'\|_2 = \|\sigma(Wx) - \sigma(W'x)\|_2 \leq \rho\|x\|_2\|W - W'\|_{\mathrm{F}}.$$

Additionally, given inputs $x$ and $x'$, we have

$$\|y - y'\|_2 = \|\sigma(Wx) - \sigma(Wx')\|_2 \leq \rho\|W\|_2\|x - x'\|_2.$$

Since vanilla RNNs are multilayer networks, Lemma 2 can be obtained by telescoping.

We remark that Lemma 2 is the key to the proof of our generalization bound, which separates the spectral norms of weight matrices and the total number of parameters.

Next, we bound the covering number of $\mathcal{F}_t$. Denote by $\mathcal{N}(\mathcal{F}_t, \epsilon, \text{dist}(\cdot, \cdot))$ the minimal cardinality of a subset $\mathcal{C} \subset \mathcal{F}_t$ that covers $\mathcal{F}_t$ at scale $\epsilon$ w.r.t the metric $\text{dist}(\cdot, \cdot)$, such that for any $f_t \in \mathcal{F}_t$, there exists $\widehat{f}_t \in \mathcal{C}$ satisfying $\text{dist}(f_t, \widehat{f}_t) = \sup_{X_t} \|f_t(X_t) - \widehat{f}_t(X_t)\|_2 \leq \epsilon$. The following lemma gives an upper bound on $\mathcal{N}(\mathcal{F}_t, \epsilon, \text{dist}(\cdot, \cdot))$.

**Lemma 3.** Under Assumptions 1–3, given any $\epsilon > 0$, the covering number of $\mathcal{F}_t$ satisfies

$$\mathcal{N}(\mathcal{F}_t, \epsilon, \text{dist}(\cdot, \cdot)) \leq \left(1 + \frac{6c\sqrt{d}t\left((\rho_h B_U)^t - 1\right)}{\epsilon\left(\rho_h B_U - 1\right)}\right)^{3d^2},$$

where $c = \rho_y \rho_h B_V B_W B_x \max\{1, \rho_h B_U\}$.

The detailed proof is provided in Appendix A.3. We briefly explain the proof technique. Given activation operators, since vanilla RNNs are in parametric forms, $f_t$ has a one-to-one correspondence to its weight matrices $U, V$, and $W$. Lemma 2 implies that $\text{dist}(\cdot, \cdot)$ is controlled by the Frobenius norms of the difference of weight matrices. Thus, it suffices to bound the covering numbers of three weight matrices, which can be obtained by the standard machinery. The product of covering numbers of three weight matrices gives us Lemma 3.

Lastly, we give an upper bound on $\mathfrak{R}_S(\mathcal{F}_{\gamma,t})$ in the following lemma.

**Lemma 4.** Under Assumptions 1–3, given activation operators and samples $S = \{(x_{i,t}, z_{i,t})_{t=1}^T, i = 1, \ldots, m\}$, the empirical Rademacher complexity $\mathfrak{R}_S(\mathcal{F}_{\gamma,t})$ satisfies

$$\mathfrak{R}_S(\mathcal{F}_{\gamma,t}) = O\left(d\rho_y B_V \min\left\{b\sqrt{d}, \rho_h B_x B_W \frac{(\rho_h B_U)^t - 1}{\rho_h B_U - 1}\right\} \frac{\sqrt{\log\left(t\sqrt{dm}\frac{(\rho_h B_U)^t - 1}{\rho_h B_U - 1}\right)}}{\sqrt{m}\gamma}\right).$$

The detailed proof is provided in Appendix A.4. Our proof exploits the Lipschitz continuity of $\mathcal{M}$ and $\ell_\gamma$, and uses Dudley's entropy integral as the standard machinery to establish Lemma 4. Combining Lemma 1 and Lemma 4, we complete the proof.

## 4 REFINED GENERALIZATION BOUNDS

When additional norm constraints on weight matrices $U, V$ and $W$ are available, we can further refine generalization bounds. Specifically, we consider the following assumptions.

**Assumption 4.** The weight matrices satisfy $\|U\|_{2,1} \leq M_U$, $\|V\|_{2,1} \leq M_V$, and $\|W\|_{2,1} \leq M_W$.

**Assumption 5.** The weight matrices satisfy $\|U\|_F \leq B_{U,F}$, $\|V\|_F \leq B_{V,F}$, and $\|W\|_F \leq B_{W,F}$.

Note that Assumption 4 appears in Bartlett et al. (2017) and Assumption 5 appears in Neyshabur et al. (2017). We have an equivalent relation between matrix norms, i.e., $\|\cdot\|_2 \leq \|\cdot\|_{2,1} \leq \sqrt{d}\|\cdot\|_F \leq d\|\cdot\|_2$. Comparing to Assumption 2, Assumptions 4 and 5 further restrict the model class. We then establish the following generalization bounds.

**Theorem 3.** Let activation operators $\sigma_h$ and $\sigma_y$ be given, and Assumptions 1–3 hold. Then for $(x_t, z_t)_{t=1}^T$ and $S = \{(x_{i,t}, z_{i,t})_{t=1}^T, i = 1, \ldots, m\}$ drawn i.i.d. from any underlying distribution over $\mathbb{R}^{d_x \times T} \times \{1, \ldots, K\}$, with probability at least $1 - \delta$ over $S$, for every margin value $\gamma > 0$ and every $f_t \in \mathcal{F}_t$ for integer $t \leq T$, the following two bounds hold:

• Suppose Assumption 4 also holds. We have

$$\mathbb{P}(\widetilde{z}_t \neq z_t) \leq \widehat{\mathcal{R}}_\gamma(f_t) + O\left(\frac{\alpha(M_U + M_V + M_W)t\frac{\beta^t - 1}{\beta - 1}\sqrt{\log d}\log(\sqrt{d}m)}{\sqrt{m}\gamma} + \sqrt{\frac{\log\frac{1}{\delta}}{m}}\right), \quad (3)$$

where $\alpha = \rho_h^2 \rho_y B_V B_W B_x$, $d = \sqrt{d_x d_h + d_h^2 + d_h d_y}$, and $\beta = \rho_h B_U$.

• Suppose Assumption 5 also holds. We have

$$\mathbb{P}(\widetilde{z}_t \neq z_t) \leq \widehat{\mathcal{R}}_\gamma(f_t) + O\left(\frac{\rho_y \rho_h (\lambda_{h_t} B_U + B_x B_W)\frac{\beta^t - 1}{\beta - 1}\sqrt{d\ln(d)(B_{U,F}^2 + B_{W,F}^2 + B_{V,F}^2)}}{\sqrt{m}\gamma}\right), \quad (4)$$

where $\lambda_t = \min\left\{b\sqrt{d}, \rho_h B_x B_W \frac{\beta^t - 1}{\beta - 1}\right\}$, $d = \sqrt{d_x d_h + d_h^2 + d_h d_y}$, and $\beta = \rho_h B_U$.

The detailed proof is provided in Appendix B.1. The first bound (3) adapts the matrix covering lemma in Bartlett et al. (2017). The second bound (4) adapts the PAC-Bayes approach (Neyshabur et al., 2017) by analyzing the divergence when imposing small perturbations on the weight matrices.

We compare generalization bounds in Table 1 by differentiating ranges[2] of $\beta$. It can be seen that when $\beta > 1$, both bounds (3) and (4) involve an exponential term in $\beta$. Thus, Theorem 2 yields a better result. When $\beta \leq 1$, we distinguish two cases: the extreme case and the approximately low rank case. Specifically, remember that in the extreme case, we have $\|\cdot\|_{2,1} = d\|\cdot\|_2$ and

---

[2]For simplicity, we assume again that $\beta$ does not scale with $t$. By $\beta < 1$, we mean $\beta \leq 1 - \Delta$ for a constant $\Delta > 1$. A similar characterization applies to $\beta > 1$.

$\| \cdot \|_{\mathrm{F}} = \sqrt{d} \| \cdot \|_2$, e.g., orthogonal weight matrices. Therefore, bound (4) meets Theorem 2 for $\beta < 1$ and is worse for $\beta = 1$. Bound (3) is worse than Theorem 2 for $\beta \leq 1$. On the other hand, if the weight matrices are approximately low rank, we have $\| \cdot \|_{2,1} \ll d \| \cdot \|_2$ and $\| \cdot \|_{\mathrm{F}} \ll \sqrt{d} \| \cdot \|_2$. In this case, bound (4) improves Theorem 2 for $\beta < 1$ by reducing dependence on $d$. Additionally, if $t(M_U + M_V + M_W) < d$, bound (3) yields tighter results both for $\beta < 1$ and $\beta = 1$. Note that $t(M_U + M_V + M_W) < d$ also implies that the input sequence is relatively short.

Table 1: Generalization bounds in Theorems 2 and 3 with respect to different ranges of $\beta$.

|  | Theorem 2 | Theorem 3 | |
| --- | --- | --- | --- |
|  |  | Bound (3) | Bound (4) |
| $\beta < 1$ | $\widetilde{O}\left(\frac{d}{\sqrt{m}\gamma}\right)$ | $\widetilde{O}\left(\frac{t(M_U+M_V+M_W)}{\sqrt{m}\gamma}\right)$ | $\widetilde{O}\left(\frac{\sqrt{d(B_{U,\mathrm{F}}^2+B_{W,\mathrm{F}}^2+B_{V,\mathrm{F}}^2)}}{\sqrt{m}\gamma}\right)$ |
| $\beta = 1$ | $\widetilde{O}\left(\frac{dt}{\sqrt{m}\gamma}\right)$ | $\widetilde{O}\left(\frac{t^2(M_U+M_V+M_W)}{\sqrt{m}\gamma}\right)$ | $\widetilde{O}\left(\frac{dt\sqrt{B_{U,\mathrm{F}}^2+B_{W,\mathrm{F}}^2+B_{V,\mathrm{F}}^2}}{\sqrt{m}\gamma}\right)$ |
| $\beta > 1$ | $\widetilde{O}\left(\frac{\sqrt{d^3 t}}{\sqrt{m}\gamma}\right)$ | $\widetilde{O}\left(\frac{t\beta^t(M_U+M_V+M_W)}{\sqrt{m}\gamma}\right)$ | $\widetilde{O}\left(\frac{d\beta^t\sqrt{B_{U,\mathrm{F}}^2+B_{W,\mathrm{F}}^2+B_{V,\mathrm{F}}^2}}{\sqrt{m}\gamma}\right)$ |

## 5 EXTENSIONS TO MGU AND LSTM RNNS

We extend our analysis to Minimal Gated Unit (MRU) and Long Short-Term Memory (LSTM) RNNs. We show that these RNNs introduce extra decaying factors to further reduce the dependence on $d$ and $t$ in generalization.

The MGU RNNs are the simplest GRU RNNs, which compute output $y_t$ as follows,

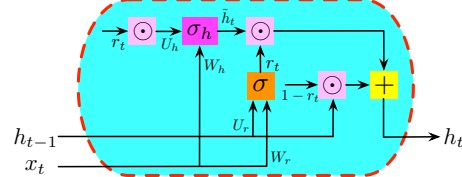

Figure 2: A basic building block of MGU RNNs

$$r_t = \sigma(W_r x_t + U_r h_{t-1}),$$
$$\widetilde{h}_t = \sigma_h\left(W_h x_t + U_h(r_t \odot h_{t-1})\right),$$
$$h_t = (1 - r_t) \odot h_{t-1} + r_t \odot \widetilde{h}_t,$$
$$y_t = \sigma_y(V h_t),$$

where $W_r, W_h \in \mathbb{R}^{d_h \times d_x}$, $U_r, U_h \in \mathbb{R}^{d_h \times d_h}$, $V \in \mathbb{R}^{d_y \times d_h}$, and $r_t \in \mathbb{R}^{d_h}$. The notation $\odot$ denotes the Hadamard product (entrywise product) of vectors. Denote by $\mathcal{F}_{g,t}$ the class of mappings from the first $t$ inputs to the $t$-th output computed by gated (MGU or LSTM) RNNs. For simplicity, we consider $\sigma$ being the sigmoid function, i.e., $\sigma(x) = (1 + \exp(-x))^{-1}$, $\sigma_h(\cdot) = \tanh(\cdot)$, and $\sigma_y$ being $\rho_y$-Lipschitz with $\sigma_y(0) = 0$. Extensions to general Lipschitz activation operators as in Assumption 3 are straightforward. Suppose we have $h_0 = 0$ and the following assumption.

**Assumption 6.** All the weight matrices have bounded spectral norms respectively, i.e. $\|W_r\|_2 \leq B_{W_r}, \|W_h\|_2 \leq B_{W_h}, \|U_r\|_2 \leq B_{U_r}, \|U_h\|_2 \leq B_{U_h}$, and $\|V\|_2 \leq B_V$.

Using a similar argument for vanilla RNNs yields a generalization bound of MGU RNNs as follows.

**Theorem 4.** Let the activation operator $\sigma_y$ be given and Assumptions 1 and 6 hold. Then for $(x_t, z_t)_{t=1}^T$ and $S = \left\{(x_{i,t}, z_{i,t})_{t=1}^T, i = 1, \ldots, m\right\}$ drawn i.i.d. from any underlying distribution over $\mathbb{R}^{d_x \times T} \times \{1, \ldots, K\}$, with probability at least $1 - \delta$ over $S$, for every margin value $\gamma > 0$ and every $f_t \in \mathcal{F}_{g,t}$ for integer $t \leq T$, we have

$$\mathbb{P}\left(\widetilde{z}_t \neq z_t\right) \leq \widehat{\mathcal{R}}_\gamma(f_t) + O\left(\frac{d\rho_y B_V \min\left\{\sqrt{d}, B_{W_h} B_x \frac{\beta^t - 1}{\beta - 1}\right\}\sqrt{\log\left(\frac{\theta^t - 1}{\theta - 1} d\sqrt{m}\right)}}{\sqrt{m}\gamma} + \sqrt{\frac{\log \frac{1}{\delta}}{m}}\right),$$

where $d = \sqrt{2d_x d_h + 2d_h^2 + d_h d_y}$, $\beta = \max_{j \leq t}\left\{\|1 - r_j\|_\infty + B_{U_h}\|r_j\|_\infty^2\right\}$, and $\theta = \beta + 2B_{U_r} + B_{U_r}B_{U_h}$.

The detailed proof is provided in Appendix C.1. As can be seen, $r_t$ shrinks the magnitude of hidden state to reduce the dependence on $d$ and $t$ in generalization. Specifically, the hidden state $h_t$ is entrywise bounded by 1. Then for any integer $t \leq T$, we have

$$\|r_t\|_\infty \leq \frac{1}{1 + \exp\left(-B_{W_r}B_x - \|U_r\|_1\right)} \quad \text{and} \quad \|1 - r_t\|_\infty \leq \frac{1}{1 + \exp\left(-B_{W_r}B_x - \|U_r\|_1\right)},$$

where $\|U_r\|_1$ denotes the maximal absolute row sum. By restricting $B_{W_r}$ and $\|U_r\|_1$ sufficiently small, we can guarantee that $\beta$ and $\theta$ are strictly smaller than 1 when $B_{U_h} = 1$ or even $B_{U_h} > 1$. As a result, with proper normalization of weight matrices, the generalization bound of MGU RNNs is less dependent on $d$ and $t$ than that of vanilla RNNs.

The LSTM RNNs are more complicated than MGU RNNs, which introduce more gates to control the information flow in RNNs. LSTM RNNs have two hidden states, and compute them as,

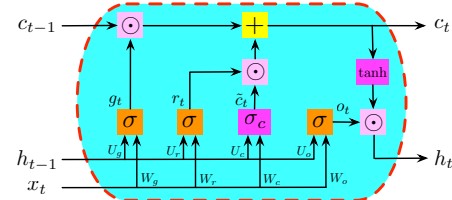

$$g_t = \sigma(W_g x_t + U_g h_{t-1}),$$
$$r_t = \sigma(W_r x_t + U_r h_{t-1}),$$
$$o_t = \sigma(W_o x_t + U_o h_{t-1}),$$
$$\widetilde{c}_t = \sigma_c \left( W_c x_t + U_c h_{t-1} \right),$$
$$c_t = g_t \odot c_{t-1} + r_t \odot \widetilde{c}_t,$$
$$h_t = o_t \odot \tanh(c_t),$$

Figure 3: A basic building block of LSTM RNNs

where $W_g, W_r, W_o, W_c \in \mathbb{R}^{d_h \times d_x}$, $U_g, U_r, U_o, U_c \in \mathbb{R}^{d_h \times d_h}$, and $g_t, r_t, o_t \in \mathbb{R}^{d_h}$. For simplicity, we also consider $\sigma$ being the sigmoid function, and $\sigma_c(\cdot) = \tanh(\cdot)$. The $t$-th output is $y_t = \sigma_y(V h_t)$, where $V \in \mathbb{R}^{d_y \times d_h}$, and $\sigma_y$ is $\rho_y$-Lipschitz with $\sigma_y(0) = 0$. Suppose we have $h_0 = c_0 = 0$ and the following assumption.

**Assumption 7.** The spectral norms of weight matrices are bounded respectively, i.e. $\|W_g\|_2 \leq B_{W_g}, \|W_r\|_2 \leq B_{W_r}, \|W_o\|_2 \leq B_{W_o}, \|W_c\|_2 \leq B_{W_c}, \|U_g\|_2 \leq B_{U_g}, \|U_r\|_2 \leq B_{U_r}, \|U_o\|_2 \leq B_{U_o}, \|U_h\|_2 \leq B_{U_h}$, and $\|V\|_2 \leq B_V$.

For properly normalized weight matrices $W_o$ and $U_o$, the generalization bound of LSTM RNNs is given in the following theorem.

**Theorem 5.** Let the activation operator $\sigma_y$ be given and Assumptions 1 and 7 hold. Then for $(x_t, z_t)_{t=1}^T$ and $S = \left\{ (x_{i,t}, z_{i,t})_{t=1}^T, i = 1, \ldots, m \right\}$ drawn i.i.d. from any underlying distribution over $\mathbb{R}^{d_x \times T} \times \{1, \ldots, K\}$, with probability at least $1 - \delta$ over $S$, for every margin value $\gamma > 0$ and every $f_t \in \mathcal{F}_{g,t}$ for integer $t \leq T$, we have

$$\mathbb{P}\left(\widetilde{z}_t \neq z_t\right) \leq \widehat{\mathcal{R}}_\gamma(f_t) + O\left( \frac{d\rho_y B_V \min\left\{\sqrt{d}, B_{W_c} B_x \frac{\beta^t - 1}{\beta - 1}\right\} \sqrt{\log\left(\frac{\theta^t - 1}{\theta - 1} d\sqrt{m}\right)}}{\sqrt{m}\gamma} + \sqrt{\frac{\log\frac{1}{\delta}}{m}} \right),$$

where $d = \sqrt{4d_x d_h + 4d_h^2 + d_h d_y}$, $\beta = \max_{j \leq t} \left\{ \|g_j\|_\infty + B_{U_c} \|r_j\|_\infty \|o_j\|_\infty \right\}$ and $\theta = \beta + B_{U_g} + B_{U_r} + B_{U_o}$.

The detailed proof is provided in Appendix C.2. Similar to MGU RNNs, LSTM RNNs also introduce extra decaying factors to reduce the dependence on $d$ and $t$ in generalization. However, LSTM RNNs are more complicated, but more flexible than MGU RNNs, since three factors, $r_t$, $o_t$ and $g_t$ are used to jointly control the spectrum of $U_c$. We further remark that LSTM RNNs also need the spectral norms of weight matrices, $W_g, W_r, W_o, U_g, U_r$, and $U_o$, to be properly controlled such that better generalization bounds can be obtained.

**Remark 1.** We also extend our analysis to convolutional RNNs (Conv RNNs, Pinheiro and Collobert (2014); Liang and Hu (2015); Xingjian et al. (2015)), and show that the convolutional filters in Conv RNNs can reduce the dependence on $d$ through parameter sharing in generalization. Due to space limit, the detailed discussion is provided in Appendix D.

## 6 NUMERICAL EVALUATION

We demonstrate a comparison among our obtained generalization bound with Bartlett et al. (2017), Neyshabur et al. (2017), and Zhang et al. (2018). Specifically, we train a vanilla RNN on the wikitext long term dependency language modeling dataset. We take $\sigma_h = \tanh$ and set the hidden state $h \in \mathbb{R}^{128}$ and the input $x \in [0, 1]^{14}$. Accordingly, we have $d = 128$ and take the sequence length $t = 56$. We list the complexity bounds for vanilla RNNs in Theorem 2 (Ours), Zhang et al. (2018) (Bound 1), (3) of Theorem 3 (Bound 2), and (4) of Theorem 3 (Bound 3) in the following by neglecting common log factors in $d$ and $t$,

- Ours: $dB_V \min\left\{\sqrt{d}, B_W \frac{B_U^t - 1}{B_U - 1}\right\} \sqrt{\log\left(\frac{B_U^t - 1}{B_U - 1}\right)}$;

- Bound 1: $dt^2 B_V B_W \max\{1, B_U^t\}$;

- Bound 2: $B_V B_W (M_U + M_V + M_W) t \frac{B_U^t - 1}{B_U - 1}$;

- Bound 3: $\left(\min\left\{\sqrt{d}, B_W \frac{B_U^t - 1}{B_U - 1}\right\} B_U + B_W\right) \frac{B_U^t - 1}{B_U - 1} \sqrt{d\left(B_{U,\mathrm{F}}^2 + B_{W,\mathrm{F}}^2 + B_{V,\mathrm{F}}^2\right)}$.

The corresponding complexity bounds are shown in Figure 4 in the logarithmic scale. As can be seen, our complexity bound in Theorem 2 is much smaller than Bounds 1-3. In more detail, the trained vanilla RNN takes $B_U = 2.6801 > 1$. As discussed earlier, for $B_U > 1$, only our bound in Theorem 2 is polynomial in the size of the network, while Bounds 1-3 are all exponential in $t$. The resulting complexity bounds corroborate such a conclusion.

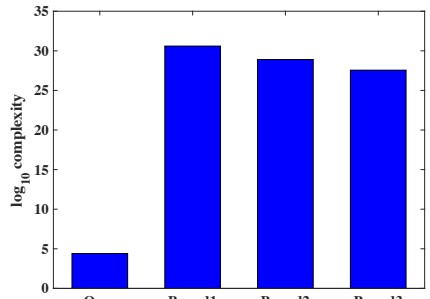

Figure 4: A comparison between generalization bounds for the same vanilla RNN trained on the wikitext dataset. The vertical axis is the logarithmic scale of the corresponding bounds.

We also observe that Bound 3 is smaller than Bound 2. The reason behind is that the weight matrices in the trained vanilla RNN have relatively small Frobenius norms but large $(2, 1)$ norms. For example, we have $B_{U,\mathrm{F}} = 13.6823$ and $M_U = 154.5439$. Then, we can calculate the stable rank $\frac{B_{U,\mathrm{F}}}{B_U} = 5.1 < \frac{\sqrt{d}}{2}$, and $\frac{M_U}{B_{U,\mathrm{F}}} = 11.3 \approx \sqrt{d}$, which implies the singular values of $U$ are not evenly distributed, while the norms of row vectors in $U$ are mostly approximately equal.

## 7 DISCUSSIONS AND OPEN PROBLEMS

**(I) Tighter bounds:** Our obtained generalization bounds depend on the spectral norms of weight matrices and the network size (the total number of parameters). Can we exploit other modeling structures to further reduce the dependence on the network size? Or can we find better choices of norms of weight matrices that yield better bounds?

**(II) Margin value:** Our generalization bounds depend on the margin value of the predictors. As can be seen, a larger margin value yields a better generalization bound. However, establishing a sharp characterization of the margin value is technically very challenging, because of its complicated dependence on the underlying data distribution and the training algorithm.

**(III) Implicit bias of SGD:** Numerous empirical evidences have already shown that RNNs trained by stochastic gradient descent (SGD) algorithms have superior generalization performance. There have been a few theoretical results showing that SGD tends to yield low complexity models, which can generalize (Neyshabur et al., 2014; 2015; Zhang et al., 2016; Soudry et al., 2017). Can we extend their argument to RNNs? For example, can SGD always yield weight matrices with well controlled spectra? As mentioned, this is crucial to the generalization of MGU and LSTM RNNs, since not well controlled spectra may even hurt generalization.

**(IV) Adaptivity to the underlying distribution:** The current PAC-Learning framework focuses on the worst case. Taking classification as an example, the theoretical analysis holds even when the input features and labels are completely independent. Therefore, this often yields very pessimistic results. For many real applications, however, data are not obtained adversarially. Some recent empirical evidences suggest that the generalization of neural networks seems very adaptive to the underlying distribution: Easier tasks lead to low complexity neural networks, while harder ones lead to highly complex neural networks. Unfortunately, none of the existing analysis can take the underlying distribution into consideration.

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

# A    PROOFS IN SECTION 2

## A.1    LIPSCHITZ CONTINUITY OF $\mathcal{M}$ AND $\ell_\gamma$

We show the Lipschitz continuity of the margin operator $\mathcal{M}$ and the loss function $\ell_\gamma$ in the following lemma.

**Lemma 5.** The margin operator $\mathcal{M}$ is 2-Lipschitz in its first argument with respect to vector Euclidean norm, and $\ell_\gamma$ is $\frac{1}{\gamma}$-Lipschitz.

*Proof.* Let $y$, $y'$ and $z$ be given, then

$$
\left| \mathcal{M}(y, z) - \mathcal{M}(y', z) \right| = \left| y_z - y'_z + \left( \max_{j \neq z} y'_j - \max_{j \neq z} y_j \right) \right|
$$
$$
\leq \left| y_z - y'_z \right| + \left| \max_{j \neq z} y'_j - y_j \right|
$$
$$
\leq 2 \left\| y - y' \right\|_\infty \leq \left\| y - y' \right\|_2 .
$$

For function $\ell_\gamma$, it is a piecewise linear function. Thus, it is straightforward to see that $\ell_\gamma$ is $\frac{1}{\gamma}$-Lipschitz.    □

## A.2    PROOF OF LEMMA 2

*Proof.* The Lemma is stated with matrix Frobenius norms. However, we can show a tighter bound only involving the spectral norms of weight matrices. Given weight matrices $U, V, W$ and $U', V', W'$, consider the $t$-th outputs $y_t$ and $y'_t$ of vanilla RNNs,

$$
\left\| y_t - y'_t \right\|_2 = \left\| \sigma_y(V h_t) - \sigma_y(V' h'_t) \right\|_2
$$
$$
\leq \rho_y \left\| V h_t - V' h_t + V' h_t - V' h'_t \right\|_2
$$
$$
\leq \rho_y \left( \left\| (V - V') h_t \right\|_2 + \left\| V'(h_t - h'_t) \right\|_2 \right)
$$
$$
\leq \rho_y \left( \left\| h_t \right\|_2 \left\| V - V' \right\|_2 + B_V \left\| h_t - h'_t \right\|_2 \right) . \tag{5}
$$

We have to bound the norm of $h_t$ as in the following lemma.

**Lemma 6.** Under Assumptions 1 to 3, for $t \geq 0$, the norm of $h_t$ is bounded by

$$
\left\| h_t \right\|_2 \leq \min \left\{ b\sqrt{d}, \rho_h B_W B_x \frac{(\rho_h B_U)^t - 1}{\rho_h B_U - 1} \right\} . \tag{6}
$$

*Proof.* We prove by induction. Observe that for $t \geq 1$, we have

$$
\left\| h_t \right\|_2 = \left\| \sigma_h(W x_t + U h_{t-1}) \right\|_2
$$
$$
\leq \rho_h \left\| W x_t + U h_{t-1} \right\|_2
$$
$$
\leq \rho_h \left( \left\| W x_t \right\|_2 + \left\| U h_{t-1} \right\|_2 \right)
$$
$$
\leq \rho_h \left( B_W B_x + B_U \left\| h_{t-1} \right\|_2 \right) . \tag{7}
$$

Applying equation (7) recursively with $h_0 = 0$, we arrive at,

$$
\left\| h_t \right\|_2 \leq \rho_h B_W B_x \sum_{j=0}^{t-1} (\rho_h B_U)^j = \rho_h B_W B_x \frac{(\rho_h B_U)^t - 1}{\rho_h B_U - 1},
$$

We also have $\left\| h_t \right\|_\infty \leq b$. Thus, combining with the above upper bound, we get $\left\| h_t \right\|_2 \leq \min \left\{ b\sqrt{d}, \rho_h B_W B_x \frac{(\rho_h B_U)^t - 1}{\rho_h B_U - 1} \right\}$. Clearly, $\left\| h_0 \right\|_2 = 0$ satisfies the upper bound.    □

When $\rho_h B_U = 1$, the ratio is defined, by L'Hospital's rule, to be the limit,

$$
\lim_{\rho_h B_U \to 1} \frac{(\rho_h B_U)^t - 1}{\rho_h B_U - 1} = t.
$$

With Lemma 6 in hand, we plug the bound (6) into equation (5) and end up with

$$\|y_t - y'_t\|_2 \le \rho_y \rho_h B_W B_x \frac{(\rho_h B_U)^t - 1}{\rho_h B_U - 1} \|V - V'\|_2 + \rho_y B_V \|h_t - h'_t\|_2 . \qquad (8)$$

The remaining task is to bound $\|h_t - h'_t\|_2$ in terms of the spectral norms of the difference of weight matrices, $\|W - W'\|_2$ and $\|U - U'\|_2$.

**Lemma 7.** Under Assumptions 1 to 3, for $t \ge 1$, the difference of hidden states $h_t$ and $h'_t$ satisfies

$$\|h_t - h'_t\|_2 \le L_{W,t} \|W - W'\|_2 + L_{U,t} \|U - U'\|_2 ,$$

where $L_{W,t} = \rho_h B_x \frac{(\rho_h B_U)^t - 1}{\rho_h B_U - 1}$ and $L_{U,t} = \rho_h^2 B_W B_x t \frac{(\rho_h B_U)_2^t - 1}{(\rho_h B_U) - 1}$.

*Proof.* Similar to the proof of Lemma 6, we use induction.

$$\begin{aligned}
\|h_t - h'_t\|_2 &= \left\|\sigma_h (W x_t + U h_{t-1}) - \sigma_h (W' x_t + U' h'_{t-1})\right\|_2 \\
&\le \rho_h \left\|(W - W') x_t + U h_{t-1} - U' h'_{t-1}\right\|_2 \\
&\le \rho_h \left(\|(W - W') x_t\|_2 + \|U h_{t-1} - U' h'_{t-1}\|_2\right) \\
&\le \rho_h \left(B_x \|W - W'\|_2 + \|U h_{t-1} - U' h_{t-1} + U' h_{t-1} - U' h'_{t-1}\|_2\right) \\
&\le \rho_h B_x \|W - W'\|_2 + \rho_h \left(\|h_{t-1}\|_2 \|U - U'\|_2 + B_U \|h_{t-1} - h'_{t-1}\|_2\right) .
\end{aligned}$$

Repeat this derivation recursively, we have

$$\begin{aligned}
\|h_t - h'_t\|_2 &\le \rho_h B_x \|W - W'\|_2 + \rho_h \|h_{t-1}\|_2 \|U - U'\|_2 + \rho_h B_U \|h_{t-1} - h'_{t-1}\|_2 \\
&\le \rho_h B_x (1 + \rho_h B_U) \|W - W'\|_2 + \rho_h (\|h_{t-1}\|_2 + \rho_h B_U \|h_{t-2}\|_2) \|U - U'\|_2 \\
&\quad + (\rho_h B_U)^2 \|h_{t-2} - h'_{t-2}\|_2 \\
&\le \cdots\cdots \\
&\le \rho_h B_x \sum_{j=0}^{t-1} (\rho_h B_U)^j \|W - W'\|_2 + \rho_h \sum_{j=0}^{t-1} \left((\rho_h B_U)^{t-1-j} \|h_j\|_2\right) \|U - U'\|_2 \\
&\quad + (\rho_h B_U)^t \|h_0 - h'_0\|_2 \\
&\le \rho_h B_x \frac{(\rho_h B_U)^t - 1}{\rho_h B_U - 1} \|W - W'\|_2 + \rho_h \sum_{j=0}^{t-1} \left((\rho_h B_U)^{t-1-j} \|h_j\|_2\right) \|U - U'\|_2 . \quad (9)
\end{aligned}$$

We now plug in the upper bound (6) to calculate the summation involving the Euclidean norms of the hidden state $h_t$.

$$\begin{aligned}
\sum_{j=0}^{t-1} (\rho_h B_U)^{t-1-j} \|h_j\|_2 &\le \sum_{j=0}^{t-1} (j+1)(\rho_h B_U)^j \rho_h B_W B_x \le t \sum_{j=0}^{t-1} (\rho_h B_U)^j \rho_h B_W B_x \\
&\le \rho_h B_W B_x t \frac{(\rho_h B_U)^t - 1}{\rho_h B_U - 1} .
\end{aligned}$$

Plugging back into equation (9), we have as desired,

$$\|h_t - h'_t\|_2 \le \rho_h B_x \frac{(\rho_h B_U)^t - 1}{\rho_h B_U - 1} \|W - W'\|_2 + \rho_h^2 B_W B_x t \frac{(\rho_h B_U)^t - 1}{\rho_h B_U - 1} \|U - U'\|_2 .$$

$\qquad\square$

Combining equation (8) and Lemma 7, and $\|W\|_F \ge \|W\|_2$, we immediately get Lemma 2. $\qquad\square$

### A.3    Proof of Lemma 3

*Proof.* Our goal is to construct a covering $\mathcal{C}(\mathcal{F}_t, \epsilon, \text{dist}(\cdot,\cdot))$, i.e., for any $f_t \in \mathcal{F}_t$, there exists $\widehat{f}_t \in \mathcal{F}_t$, for any input data $(x_t)_{t=1}^T$, satisfying

$$\sup_{X_t} \left\| f_t(X_t) - \widehat{f}_t(X_t) \right\|_2 \leq \epsilon.$$

Note that $f$ is determined by weight matrices $U, V$ and $W$. By Lemma 2, we have

$$\sup_{X_t} \left\| f(X_t) - \widehat{f}(X_t) \right\|_2 \leq L_{V,t} \left\| V - \widehat{V} \right\|_F + L_{W,t} \left\| W - \widehat{W} \right\|_F + L_{U,t} \left\| U - \widehat{U} \right\|_F.$$

Then it is enough to construct three matrix coverings, $\mathcal{C}\left(U, \frac{\epsilon}{3L_{U,t}}, \|\cdot\|_F\right)$, $\mathcal{C}\left(V, \frac{\epsilon}{3L_{V,t}}, \|\cdot\|_F\right)$ and $\mathcal{C}\left(W, \frac{\epsilon}{3L_{W,t}}, \|\cdot\|_F\right)$. Their Cartesian product gives us the covering $\mathcal{C}(\mathcal{F}_t, \epsilon, \text{dist}(\cdot,\cdot))$. The following lemma gives an upper bound on the covering number of matrices with a bounded Frobenius norm.

**Lemma 8.** Let $\mathcal{G} = \left\{ A \in \mathbb{R}^{d_1 \times d_2} : \|A\|_2 \leq \lambda \right\}$ be the set of matrices with bounded spectral norm and $\epsilon > 0$ be given. The covering number $\mathcal{N}(\mathcal{G}, \epsilon, \|\cdot\|_F)$ is upper bounded by

$$\mathcal{N}(\mathcal{G}, \epsilon, \|\cdot\|_F) \leq \left( 1 + 2 \frac{\min\left\{ \sqrt{d_1}, \sqrt{d_2} \right\} \lambda}{\epsilon} \right)^{d_1 d_2}.$$

*Proof.* For any matrix $A \in \mathcal{G}$, we define a mapping $\phi : \mathbb{R}^{d_1 \times d_2} \mapsto \mathbb{R}^{d_1 d_2}$, such that $\phi(A) = [A_{:,1}^\top, A_{:,2}^\top, \ldots, A_{:,h}^\top]^\top$, where $A_{:,i}$ denotes the $i$-th column of matrix $A$. Denote the vector space induced by the mapping $\phi$ by $\mathcal{V}(\mathcal{G}) = \{\phi(A) : A \in \mathcal{G}\}$. Note that we have $\|A\|_F^2 = \sum_{i=1}^h A_{:,i}^\top A_{:,i} = \|\phi(A)\|_2^2$ and the mapping $\phi$ is one-to-one and onto. By definition, the square of Frobenius norm equals the square of sum of singular values and the spectral norm is the largest singular value. Hence, the equivalence of Frobenius norm and spectral norm is given by the following inequalities,

$$\|A\|_2 \leq \|A\|_F \leq \min\left\{ \sqrt{d_1}, \sqrt{d_2} \right\} \|A\|_2.$$

Now, we see that if we construct a covering $\mathcal{C}(\mathcal{V}(\mathcal{G}), \epsilon, \|\cdot\|_2)$, then $\phi^{-1}\mathcal{C}(\mathcal{V}(\mathcal{G}), \epsilon, \|\cdot\|_2) = \left\{ \phi^{-1}(v) : v \in \mathcal{C}(\mathcal{V}(\mathcal{G}), \epsilon, \|\cdot\|_2) \right\}$ is a covering of $\mathcal{G}$ at scale $\epsilon$ with respect to the matrix Frobenius norm. Therefore, we get $\mathcal{N}(\mathcal{G}, \epsilon, \|\cdot\|_F) \leq \mathcal{N}(\mathcal{V}(\mathcal{G}), \epsilon, \|\cdot\|_2)$. As a consequence, it is suffices to upper bound the covering number of $\mathcal{V}(\mathcal{G})$. In order to do so, we need another closely related concept, packing number.

**Definition 2** (Packing). Let $\mathcal{G}$ be an arbitrary set and $\epsilon > 0$ be given. We say $\mathcal{P}(\mathcal{G}, \epsilon, \|\cdot\|)$ is a packing of $\mathcal{G}$ at scale $\epsilon$ with respect to the norm $\|\cdot\|$, if for any two elements $A, B \in \mathcal{P}$, we have

$$\|A - B\| > \epsilon.$$

Denote by $\mathcal{M}(\mathcal{G}, \epsilon, \|\cdot\|)$ the maximal cardinality of $\mathcal{P}(\mathcal{G}, \epsilon, \|\cdot\|)$.

By the maximality, we can check that $\mathcal{N}(C, \epsilon, \|\cdot\|) \leq \mathcal{M}(C, \epsilon, \|\cdot\|)$. Indeed, let $\mathcal{P}^*(\mathcal{G}, \epsilon, \|\cdot\|)$ be a maximal packing. Suppose there exists $A \in \mathcal{G}$ such that for any $B \in \mathcal{P}^*(\mathcal{G}, \epsilon, \|\cdot\|)$, the inequality $\|A - B\| > \epsilon$ holds. Then we can add $A$ to $\mathcal{P}^*(\mathcal{G}, \epsilon, \|\cdot\|)$, while still keeping it being a packing, which contradicts the maximality of $\mathcal{P}^*(\mathcal{G}, \epsilon, \|\cdot\|)$. Thus, we have $\mathcal{N}(\mathcal{G}, \epsilon, \|\cdot\|) \leq \mathcal{M}(\mathcal{G}, \epsilon, \|\cdot\|)$.

Observe that $\mathcal{V}(\mathcal{G})$ is contained in an Euclidean ball $\mathcal{B}(0; R) \in \mathbb{R}^{d_1 d_2}$ of radius at most

$$R = \max_{A \in \mathcal{G}} \|\phi(A)\|_2 \leq \min\left\{ \sqrt{d_1}, \sqrt{d_2} \right\} \|A\|_2 \leq \min\left\{ \sqrt{d_1}, \sqrt{d_2} \right\} \lambda.$$

Additionally, the union of Euclidean balls $\mathcal{B}(v; \epsilon/2) \subset \mathbb{R}^{d_1 d_2}$ with radius $\epsilon/2$ and center $v \in \mathcal{P}(\mathcal{V}(\mathcal{G}), \epsilon, \|\cdot\|_2)$ is further contained in an Euclidean ball $\mathcal{B}(0; R_\epsilon)$ of slightly enlarged radius $R_\epsilon = \min\left\{ \sqrt{d_1}, \sqrt{d_2} \right\} \lambda + \epsilon/2$. Those balls $\mathcal{B}(v; \epsilon/2)$ are disjoint by the definition of packing, thus we have

$$\mathcal{N}(\mathcal{V}(C), \epsilon, \|\cdot\|_2) \leq \mathcal{P}(\mathcal{V}(C), \epsilon, \|\cdot\|_2) \leq \frac{\text{vol}(\mathcal{B}(0, R_\epsilon))}{\text{vol}(\mathcal{B}(v; \epsilon/2))} = \left( \frac{R_\epsilon}{\epsilon/2} \right)^{d_1 d_2}$$

$$= \left( 1 + 2 \frac{\min\{\sqrt{d_1}, \sqrt{d_2}\}\lambda}{\epsilon} \right)^{d_1 d_2},$$

where $\text{vol}(\cdot)$ denotes the volume. $\qquad\qquad\square$

By Lemma 8, we can directly write out the upper bounds on the covering numbers of weight matrices,

$$\mathcal{N}\left(U, \frac{\epsilon}{3L_{U,t}}, \|\cdot\|_{\mathrm{F}}\right) \le \left(1 + 6\frac{\sqrt{d_h}B_U L_{U,t}}{\epsilon}\right)^{d_h^2},$$

$$\mathcal{N}\left(V, \frac{\epsilon}{3L_{V,t}}, \|\cdot\|_{\mathrm{F}}\right) \le \left(1 + 6\frac{\min\{\sqrt{d_y}, \sqrt{d_h}\}B_V L_{V,t}}{\epsilon}\right)^{d_y d_h}, \quad \text{and}$$

$$\mathcal{N}\left(W, \frac{\epsilon}{3L_{W,t}}, \|\cdot\|_{\mathrm{F}}\right) \le \left(1 + 6\frac{\min\{\sqrt{d_x}, \sqrt{d_h}\}B_W L_{W,t}}{\epsilon}\right)^{d_x d_h}.$$

Then we immediately have,

$$\mathcal{N}(\mathcal{F}_t, \epsilon, \mathrm{dist}(\cdot, \cdot)) \le \mathcal{N}\left(U, \frac{\epsilon}{3L_{U,t}}, \|\cdot\|_{\mathrm{F}}\right) \times \mathcal{N}\left(V, \frac{\epsilon}{3L_{V,t}}, \|\cdot\|_{\mathrm{F}}\right) \times \mathcal{N}\left(W, \frac{\epsilon}{3L_{W,t}}, \|\cdot\|_{\mathrm{F}}\right)$$

$$\le \left(1 + \frac{6\sqrt{d_h}B_U L_{U,t}}{\epsilon}\right)^{d_h^2} \left(1 + \frac{\min\{6\sqrt{d_y}, \sqrt{d_h}\}B_V L_{V,t}}{\epsilon}\right)^{d_y d_h}$$

$$\times \left(1 + \frac{6\min\{\sqrt{d_x}, \sqrt{d_h}\}B_W L_{W,t}}{\epsilon}\right)^{d_x d_h}.$$

Substituting the coefficients $L_{U,t}$, $L_{V,t}$ and $L_{W,t}$ from Lemma 2, we get

$$\mathcal{N}(\mathcal{F}_t, \epsilon, \mathrm{dist}(\cdot, \cdot))$$

$$\le \left(1 + \frac{6\sqrt{d}\rho_y\rho_h B_V B_W B_x \frac{(\rho_h B_U)^t - 1}{\rho_h B_U - 1}}{\epsilon}\right)^{2d^2} \left(1 + \frac{6\sqrt{d}\rho_y\rho_h^2 B_U B_V B_W B_x t \frac{(\rho_h B_U)^t - 1}{\rho_h B_U - 1}}{\epsilon}\right)^{d^2}$$

$$\le \left(1 + \frac{6c\sqrt{d}t \frac{(\rho_h B_U)^t - 1}{\rho_h B_U - 1}}{\epsilon}\right)^{3d^2},$$

where $c = \rho_y\rho_h B_V B_W B_x \max\{1, \rho_h B_U\}$. For future usage, we also write down for small $\epsilon > 0$, such that $\frac{6c\sqrt{d}t \frac{(\rho_h B_U)^t - 1}{\rho_h B_U - 1}}{\epsilon} > 1$, the logarithm of covering number satisfies,

$$\log\mathcal{N}(\mathcal{F}_t, \epsilon, \mathrm{dist}(\cdot, \cdot)) \le 3d^2 \log\left(\frac{12c\sqrt{d}t \frac{(\rho_h B_U)^t - 1}{\rho_h B_U - 1}}{\epsilon}\right).$$

$\square$

### A.4 PROOF OF LEMMA 4

*Proof.* Define $\mathcal{F}_{\mathcal{M},t} = \{(X_t, z_t) \mapsto \mathcal{M}(f_t(X_t), z_t) : f_t \in \mathcal{F}_t\}$. By Lemma 5, we see that $\mathcal{M}$ is 2-Lipschitz in its first argument. In order to cover $\mathcal{F}_{\mathcal{M},t}$ at scale $\epsilon$, it suffices to cover $\mathcal{F}_t$ at scale $\frac{\epsilon}{2}$. This immediately gives us the covering number $\mathcal{N}(\mathcal{F}_{\mathcal{M},t}, \epsilon, \|\cdot\|_\infty) \le \mathcal{N}(\mathcal{F}_t, \epsilon/2, \mathrm{dist}(\cdot, \cdot))$.

We then give the statement of Dudley's entropy integral.

**Lemma 9.** Let $\mathcal{H}$ be a real-valued function class taking values in $[-r, r]$ for some constant $r$, and assume that $0 \in \mathcal{H}$. Let $S = (s_1, \ldots, s_m)$ be given points, then

$$\Re_S(\mathcal{H}) \le \inf_{\alpha > 0}\left(\frac{4\alpha}{\sqrt{m}} + \frac{12}{m}\int_\alpha^{2r\sqrt{m}} \sqrt{\log\mathcal{N}(\mathcal{H}, \epsilon, \|\cdot\|)}d\epsilon\right).$$

The proof can be found in Bartlett et al. (2017). Taking $\mathcal{H} = \mathcal{F}_{\mathcal{M},t}$, we can easily verify that $\mathcal{F}_{\mathcal{M},t}$ takes values in $[-r, r]$ with $r = \rho_y B_V \|h_t\|_2 \le \rho_y B_V \min\left\{b\sqrt{d}, \rho_h B_W B_x \frac{(\rho_h B_U)^t - 1}{\rho_h B_U - 1}\right\}$ and

$0 \in \mathcal{F}_\mathcal{M}$. Thus, directly applying Lemma 9 yields the following bound,

$$\mathfrak{R}_S(\mathcal{F}_{\mathcal{M},t}) \leq \inf_{\alpha>0} \left( \frac{4\alpha}{\sqrt{m}} + \frac{12}{m} \int_\alpha^{2r\sqrt{m}} \sqrt{\log \mathcal{N}(\mathcal{F}_{\mathcal{M},t}, \epsilon, \|\cdot\|_\infty)} d\epsilon \right).$$

We bound the integral as follows,

$$\int_\alpha^{2r\sqrt{m}} \sqrt{\log \mathcal{N}(\mathcal{F}_{\mathcal{M},t}, \epsilon, \|\cdot\|_\infty)} d\epsilon \leq \int_\alpha^{2r\sqrt{m}} \sqrt{3d^2 \log \left( \frac{24c\sqrt{d}t \frac{(\rho_h B_U)^t - 1}{\rho_h B_U - 1}}{\epsilon} \right)} d\epsilon$$

$$\leq 2r\sqrt{m} \sqrt{3d^2 \log \left( \frac{24c\sqrt{d}t \frac{(\rho_h B_U)^t - 1}{\rho_h B_U - 1}}{\alpha} \right)}.$$

Picking $\alpha = \frac{1}{\sqrt{m}}$ is enough to give us an upper bound on $\mathfrak{R}_S(\mathcal{F}_{\mathcal{M},t})$,

$$\mathfrak{R}_S(\mathcal{F}_\mathcal{M}) \leq \frac{4}{m} + \frac{24}{\sqrt{m}} \sqrt{3d^2 r^2 \log \left( 24c\sqrt{d}mt \frac{(\rho_h B_U)^t - 1}{\rho_h B_U - 1} \right)}.$$

Finally, by Talagrand's lemma (Mohri et al., 2012) and $\ell_\gamma$ being $\frac{1}{\gamma}$-Lipschitz, we have

$$\mathfrak{R}_S(\mathcal{F}_{\gamma,t}) \leq \frac{1}{\gamma} \mathfrak{R}_S(\mathcal{F}_{\mathcal{M},t}) \leq \frac{4}{m\gamma} + \frac{24}{\sqrt{m}\gamma} \sqrt{3d^2 r^2 \log \left( 24c\sqrt{d}mt \frac{(\rho_h B_U)^t - 1}{\rho_h B_U - 1} \right)}.$$

$\square$

# B PROOF IN SECTION 4

## B.1 PROOF OF THEOREM 3

*Proof.* Under additional Assumption 4, we only need to show that, with the additional matrix induced norm bound, we have a refined upper bound on the matrix covering number. The proof relies on the following lemma adapted from Bartlett et al. (2017) Lemma 3.2.

**Lemma 10.** Let $\mathcal{G} = \{A \in \mathbb{R}^{d_1 \times d_2} : \|A\|_{2,1} \leq \lambda\}$. We have the following matrix covering upper bound

$$\log \mathcal{N}(\mathcal{G}, \epsilon, \|\cdot\|_2) \leq \frac{\lambda^2}{\epsilon^2} \log(2d_1 d_2).$$

The above Lemma is a direct consequence of Lemma 3.2 in Bartlett et al. (2017) with $X$ being identity, $a = \lambda$, $b = 1$, and $m = d_1, d = d_2$. We apply the same trick to split the overall covering accuracy $\epsilon$ into 3 parts, $\frac{\epsilon}{3L_{U,t}}$, $\frac{\epsilon}{3L_{V,t}}$, and $\frac{\epsilon}{3L_{W,t}}$, corresponding to $U, V, W$ respectively. Then we derive a refined bound on the covering number of $\mathcal{F}_t$:

$$\log \mathcal{N}(\mathcal{F}_t, \epsilon, \text{dist}(\cdot, \cdot)) \leq \frac{9 \left( M_U L_{U,t}^2 + M_V L_{V,t}^2 + M_W L_{W,t}^2 \right)}{\epsilon^2} \log(2d^2), \tag{10}$$

where $d = \max \{d_x, d_y, d_h\}$. Substituting (10) into the Dudley integral as in the proof of Lemma 4 yields

$$\mathfrak{R}_S(\mathcal{F}_{\mathcal{M},t}) \leq \inf_{\alpha>0} \left( \frac{4\alpha}{\sqrt{m}} + \frac{12}{m} \int_\alpha^{2r\sqrt{m}} \sqrt{\log \mathcal{N}(\mathcal{F}_t, \epsilon/2, \|\cdot\|_\infty)} d\epsilon \right).$$

We bound the integral as follows,

$$\int_\alpha^{2r\sqrt{m}} \sqrt{\log \mathcal{N}(\mathcal{F}_t, \epsilon/2, \|\cdot\|_\infty)} d\epsilon \leq \int_\alpha^{2r\sqrt{m}} 36 \frac{\sqrt{M_U L_{U,t}^2 + M_V L_{V,t}^2 + M_W L_{W,t}^2}}{\epsilon} \sqrt{\log(2d^2)} d\epsilon$$

$$= 36 \sqrt{M_U L_{U,t}^2 + M_V L_{V,t}^2 + M_W L_{W,t}^2} \sqrt{\log(2d^2)} \log \frac{2r\sqrt{m}}{\alpha}.$$

Choosing $\alpha = \frac{1}{\sqrt{m}}$ yields

$$\mathfrak{R}_S(\mathcal{F}_{\mathcal{M}}) \leq \frac{4}{m} + \frac{432}{\sqrt{m}}\sqrt{M_U L_{U,t}^2 + M_V L_{V,t}^2 + M_W L_{W,t}^2}\sqrt{\log(2d^2)}\log\left(2m\sqrt{d}\right).$$

Finally, substituting the Lipschitz constant $L_{U,t}, L_{V,t}, L_{W,t}$ into the expression, we have

$$\mathfrak{R}_S(\mathcal{F}_{\gamma,t}) \leq \frac{1}{\gamma}\mathfrak{R}_S(\mathcal{F}_{\mathcal{M},t}) \leq \frac{4}{m\gamma} + \frac{432}{\gamma\sqrt{m}}\sqrt{M_U L_{U,t}^2 + M_V L_{V,t}^2 + M_W L_{W,t}^2}\sqrt{\log(2d^2)}\log\left(2m\sqrt{d}\right)$$

$$\leq O\left(\frac{\alpha\max\{M_U, M_V, M_W\}t\frac{\beta^t-1}{\beta-1}}{\gamma\sqrt{m}}\sqrt{\log d}\log\left(m\sqrt{d}\right)\right).$$

Combining with Lemma 1 completes the proof.

Under additional Assumption 5, our proof is based on the following result from Lemma 1 in Neyshabur et al. (2017).

**Lemma 11.** Let $f_\alpha(x) : \mathcal{X} \to \mathbb{R}^d$ be any predictor with parameter $\alpha$, and $\mathcal{P}$ be any distribution on the parameter that is independent of training data. Then, for any $\gamma, \delta > 0$, with probability at least $1 - \delta$ over the training set of size $m$, for any $\alpha$ and any random perturbation $\beta$ s.t. $\mathbb{P}_\beta\left[\max_{x\in\mathcal{X}}|f_{\alpha+\beta}(x) - f_\alpha(x)|_\infty < \frac{\gamma}{4}\right] \geq \frac{1}{2}$, we have

$$\mathcal{R}_0(f_\alpha) - \widehat{\mathcal{R}}_\gamma(f_\alpha) \leq 4\sqrt{\frac{\mathrm{KL}(\alpha+\beta\|\mathcal{P}) + \log\left(\frac{6m}{\delta}\right)}{m-1}},$$

where $\mathrm{KL}(\alpha+\beta\|\mathcal{P})$ is KL divergence of distributions $\alpha+\beta$ and $\mathcal{P}$.

For convenience, we omit the superscript for sample index. Denote $h_t(\alpha)$ and $h_t(\alpha+\beta)$ as the hidden variables with parameters $\alpha$ and $\alpha+\beta$ respectively. Then we provide an upper bound of the gap of hidden layers before and after the perturbation. Denote the parameters $\alpha = \mathrm{vec}(\{W, U, V\})$ and the perturbation $\beta = \mathrm{vec}(\{\delta W, \delta U, \delta V\})$.

For any $t \in \{1, 2, \ldots, T\}$, we have

$\|h_t(\alpha+\beta) - h_t(\alpha)\|_2$

$\overset{(i)}{\leq} \rho_h\|(U+\delta U)h_{t-1}(\alpha+\beta) + (W+\delta W)x_t - Uh_{t-1}(\alpha) - Wx_t\|_2$

$\overset{(ii)}{\leq} \rho_h B_U\|h_{t-1}(\alpha+\beta) - h_{t-1}(\alpha)\|_2 + \delta\rho_h B_U\|h_{t-1}(\alpha+\beta)\|_2 + \delta\rho_h B_x B_W$

$\leq (\rho_h B_U)^t\|h_0(\alpha+\beta) - h_0(\alpha)\|_2 + \delta\sum_{i=1}^{t}(\rho_h B_U)^i\|h_{(t-i)}(\alpha+\beta)\|_2 + \delta\rho_h B_x B_W\sum_{i=0}^{t-1}(\rho_h B_U)^i,$ (11)

By Lemma 6, we have that for any $t \leq T$,

$$\|h_t(\alpha)\|_2 \leq \min\left\{b\sqrt{p}, \rho_h B_x B_W\frac{(\rho_h B_U)^t-1}{\rho_h B_U-1}\right\} = \lambda_{h_t}. \tag{12}$$

Combining (11), (12), and $h_{(0)} = 0$, we have

$$\left\|h_{(t)}(\alpha+\beta) - h_{(t)}(\alpha)\right\|_2 \leq \delta\lambda_{h_t}\sum_{i=1}^{t}(\rho_h B_U)^i + \delta\rho_h B_x B_W\sum_{i=0}^{t-1}(\rho_h B_U)^i$$

$$\leq \delta(\lambda_{h_t}\rho_h B_U + \rho_h B_x B_W)\frac{(\rho_h B_U)^t-1}{\rho_h B_U-1}. \tag{13}$$

Denote $y_t(\alpha)$ and $y_t(\alpha+\beta)$ as the out with parameters $\alpha$ and $\alpha+\beta$ respectively. Then we have

$\left\|y_{(t)}(\alpha+\beta) - y_{(t)}(\alpha)\right\|_2 \overset{(i)}{\leq} \rho_y\|(1+\delta)Vh_t(\alpha+\beta) - Vh_t(\alpha)\|_2$

$\leq \rho_y B_V\|h_t(\alpha+\beta) - h_t(\alpha)\|_2 + \delta\rho_y B_V\|h_t(\alpha+\beta)\|_2$

$\overset{(ii)}{\leq} \delta\rho_y B_V(\lambda_{h_t}\rho_h B_U + \rho_h B_x B_W)\frac{(\rho_h B_U)^t-1}{\rho_h B_U-1} + \delta\rho_y B_V\lambda_{h_t},$ (14)

where $(i)$ is from Lipschitz continuity of $\sigma_y$ and $(ii)$ is from (12) and (13).

Then choosing the prior distribution and the perturbation distribution as $\mathcal{N}\left(0, \sigma^2 I\right)$, and from the concentration result for the spectral norm bounds, we have

$$\mathbb{P}_{A \sim \mathcal{N}(0, \sigma^2 I_{d \times d})}\left[\|A\|_2 > \xi\right] \leq 2p \exp\left(\frac{-\xi^2}{2d\sigma^2}\right).$$

This implies with probability at least $1/2$, we have $\max\{\delta B_U, \delta B_W, \delta B_V\} \leq \sigma\sqrt{2d\ln(12d)}$. Taking $\sigma = \left(\gamma/4\rho_y\left(\left(\lambda_{h_t}\rho_h B_U + \rho_h B_x B_W\right)\frac{(\rho_h B_U)^t - 1}{\rho_h B_U - 1} + \lambda_{h_t}\right)\sqrt{2d\ln(12d)}\right)$ and combining with (14), with probability at least $1/2$, we have

$$\max_{x \in \mathcal{X}_m}\|y_t\left(\alpha + \beta\right) - y_t\left(\alpha\right)\|_2$$
$$\leq \left(\left(\lambda_{h_t}\rho_h B_U + \rho_h B_x B_W\right)\frac{(\rho_h B_U)^t - 1}{\rho_h B_U - 1} + \lambda_{h_t}\right) \cdot \sigma\sqrt{2d\ln(12d)} \leq \frac{\gamma}{4}.$$

Finally, we calculate the KL divergence of $\mathcal{P}$ and $\alpha + \beta$ with respect to this choice of $\sigma$,

$$\mathrm{KL}\left(\alpha + \beta \| \mathcal{P}\right) \leq \frac{\|\alpha\|_2^2}{2\sigma^2}$$
$$= O\left(\frac{\rho_y^2}{\gamma^2} \cdot \left(\left(\lambda_{h_t}\rho_h B_U + \rho_h B_x B_W\right)\frac{(\rho_h B_U)^t - 1}{\rho_h B_U - 1} + \lambda_{h_t}\right)^2 d\ln(d)\left(B_{U,\mathrm{F}}^2 + B_{W,\mathrm{F}}^2 + B_{V,\mathrm{F}}^2\right)\right)$$
$$= O\left(\frac{\rho_y^2\left(\lambda_{h_t}\rho_h B_U + \rho_h B_x B_W\right)^2\left(\beta^t - 1\right)^2 p\ln(p)\left(B_{U,\mathrm{F}}^2 + B_{W,\mathrm{F}}^2 + B_{V,\mathrm{F}}^2\right)}{\gamma^2\left(\beta - 1\right)^2}\right).$$

We complete the proof by applying Lemma 11. $\qquad\square$

## C  Proofs in Section 5

### C.1  Proof of Theorem 4

*Proof.* We use the same argument from the analysis of vanilla RNNs to investigate the Lipschitz continuity of MGU RNNs. Consider $h_t$ and $h_t'$ computed by different sets of weight matrices.

$$\|h_t - h_t'\|_2 = \left\|(1 - r_t) \odot h_{t-1} + r_t \odot \widetilde{h}_t - (1 - r_t') \odot h_{t-1}' - r_t' \odot \widetilde{h}_t'\right\|_2$$
$$\leq \left\|(r_t' - r_t) \odot h_{t-1}'\right\|_2 + \left\|(1 - r_t) \odot (h_{t-1} - h_{t-1}')\right\|_2 + \left\|(r_t - r_t') \odot \widetilde{h}_t'\right\|_2 + \left\|r_t \odot (\widetilde{h}_t - \widetilde{h}_t')\right\|_2$$
$$\leq \|r_t' - r_t\|_2\|h_{t-1}'\|_\infty + \|1 - r_t\|_\infty\|h_{t-1} - h_{t-1}'\|_2 + \|r_t - r_t'\|_2\left\|\widetilde{h}_t'\right\|_\infty + \|r_t\|_\infty\left\|\widetilde{h}_t - \widetilde{h}_t'\right\|_2$$

Expand the expression of $\widetilde{h}_t$. Note that $r_t$ is nonnegative, and $\|r_t\|_\infty \leq 1$. Then we have $\|h_t\|_\infty \leq 1$. Additionally $\tanh(\cdot)$ is 1-Lipschitz. Thus we get

$$\left\|\widetilde{h}_t - \widetilde{h}_t'\right\|_2 \leq \left\|U_h(h_{t-1} \odot r_t) + W_h x_t - U_h'(h_{t-1}' \odot r_t') - W_h' x_t\right\|_2$$
$$\leq \left\|U_h(h_{t-1} \odot r_t) - U_h'(h_{t-1}' \odot r_t')\right\|_2 + B_x\|W_h - W_h'\|_2$$
$$\leq \|U_h - U_h'\|_2\|h_{t-1} \odot r_t\|_2 + B_{U_h}\|r_t\|_\infty\|h_{t-1} - h_{t-1}'\|_2 + B_{U_h}\|h_{t-1}'\|_\infty\|r_t - r_t'\|_2$$
$$\quad + B_x\|W_h - W_h'\|_2$$
$$\leq \|h_t\|_2\|U_h - U_h'\|_2 + B_{U_h}\|r_t\|_\infty\|h_{t-1} - h_{t-1}'\|_2 + B_{U_h}\|r_t - r_t'\|_2 + B_x\|W_h - W_h'\|_2.$$

We have to expand $r_t - r_t'$ as follows,

$$\|r_t - r_t'\|_2 = \left\|W_r x_t + U_r h_{t-1} - W_r' x_t - U_r' h_{t-1}'\right\|_2$$
$$\leq B_x\|W_r - W_r'\|_2 + B_{U_r}\left\|h_{t-1} - h_{t-1}'\right\|_2 + \|h_{t-1}'\|_2\|U_r - U_r'\|_2.$$

We also need to bound $\|h_t\|_2$,

$$
\begin{aligned}
\|h_t\|_2 &\leq \|1 - r_t\|_\infty \|h_{t-1}\|_2 + \|r_t\|_\infty \left\|\widetilde{h}_t\right\|_2 \\
&\leq \|1 - r_t\|_\infty \|h_{t-1}\|_2 + \|r_t\|_\infty \left(B_{W_h} B_x + B_{U_h} \|r_t\|_\infty \|h_{t-1}\|_2\right) \\
&= \left(\|1 - r_t\|_\infty + B_{U_h} \|r_t\|_\infty^2\right) \|h_{t-1}\|_2 + B_{W_h} B_x, \\
&\leq \max_{j \leq t} \left\{\|1 - r_j\|_\infty + B_{U_h} \|r_j\|_\infty^2\right\} \|h_{t-1}\|_2 + B_{W_h} B_x.
\end{aligned}
$$

Applying the above inequality recursively and remember $\|h_t\|_\infty \leq 1$, we get $\|h_t\|_2 \leq \min\left\{\sqrt{d}, \frac{\beta^t - 1}{\beta - 1} B_{W_h} B_x\right\}$ with $\beta = \max_{j \leq t}\left\{\|1 - r_j\|_\infty + B_{U_h} \|r_j\|_\infty^2\right\}$. Put all the above ingredients together, we have

$$
\begin{aligned}
\|h_t - h_t'\|_2 &\leq (\beta + 2B_{U_r} + B_{U_r} B_{U_h}) \left\|h_{t-1} - h_{t-1}'\right\|_2 \\
&\quad + \sqrt{d} \|U_h - U_h'\|_2 + B_x \|W_h - W_h'\|_2 \\
&\quad + (2 + B_{U_h}) \sqrt{d} \|U_r - U_r'\|_2 + (2B_x + B_{U_h} B_x) \|W_r - W_r'\|_2.
\end{aligned}
$$

Apply the above inequality recursively, denote by $\theta = \beta + 2B_{U_r} + B_{U_r} B_{U_h}$, we have

$$
\begin{aligned}
\|h_t - h_t'\|_2 \leq &\sqrt{d} \sum_{j=1}^t \theta^j \|U_h - U_h'\|_2 + B_x \sum_{j=1}^t \theta^j \|W_h - W_h'\|_2 \\
&+ \left(2\sqrt{d} + B_{U_h}\sqrt{d}\right) \sum_{j=1}^t \theta^j \|U_r - U_r'\|_2 + (2B_x + B_{U_h} B_x) \sum_{j=1}^t \theta^j \|W_r - W_r'\|_2.
\end{aligned}
$$

We then derive the Lipschitz continuity of $\|y_t\|_2$,

$$
\begin{aligned}
&\|y_t - y_t'\|_2 \leq \rho_y B_V \|h_t - h_t'\|_2 + \rho_y \sqrt{d}\|V - V'\|_2 \\
&\leq \rho_y B_V \sqrt{d} \frac{\theta^t - 1}{\theta - 1} \|U_h - U_h'\|_2 + \rho_y B_V B_x \frac{\theta^t - 1}{\theta - 1} \|W_h - W_h'\|_2 + \rho_y \sqrt{d}\|V - V'\|_2 \\
&+ \rho_y B_V \left(2\sqrt{d} + B_{U_h}\sqrt{d}\right) \frac{\theta^t - 1}{\theta - 1} \|U_r - U_r'\|_2 + \rho_y B_V (2B_x + B_{U_h} B_x) \frac{\theta^t - 1}{\theta - 1} \|W_r - W_r'\|_2.
\end{aligned}
$$

Following the same argument for proving the generalization bound of vanilla RNNs, we can get the generalization bound for MGU RNNs as

$$
\mathbb{P}\left(\widetilde{z}_t \neq z_t\right) \leq \widehat{\mathcal{R}}_\gamma(f_t) + O\left(\frac{d\rho_y B_V \min\left\{\sqrt{d}, B_{W_h} B_x \frac{\beta^t - 1}{\beta - 1}\right\} \sqrt{\log\left(d\sqrt{m}\frac{\theta^t - 1}{\theta - 1}\right)}}{\sqrt{m}\gamma} + \sqrt{\frac{\log\frac{1}{\delta}}{m}}\right).
$$

$\square$

### C.2 Proof of Theorem 5

*Proof.* We first bound the norm of $h_t$ as follows,

$$
\begin{aligned}
\|h_t\|_2 &\leq \|o_t\|_\infty \|\tanh(c_t)\|_2 \leq \|o_t\|_\infty \|c_t\|_2 \\
&\leq \|g_t\|_\infty \|c_{t-1}\|_2 + \|r_t\|_\infty \|\widetilde{c}_t\|_2 \\
&\leq \|g_t\|_\infty \|c_{t-1}\|_2 + \|r_t\|_\infty \left(B_{W_c} B_x + B_{U_c} \|h_{t-1}\|_2\right) \\
&\leq \|g_t\|_\infty \|c_{t-1}\|_2 + \|r_t\|_\infty \left(B_{W_c} B_x + B_{U_c} \|o_t\|_\infty \|c_{t-1}\|_2\right) \\
&\leq \left(\|g_t\|_\infty + \|r_t\|_\infty \|o_t\|_\infty B_{U_c}\right) \|c_{t-1}\|_2 + B_{W_c} B_x.
\end{aligned}
$$

By applying the above inequality recursively, we have $\|h_t\|_2 \leq \|c_t\|_2 \leq B_{W_c} B_x \frac{\beta^t - 1}{\beta^t - 1}$, where $\beta = \max_{j \leq t} \{\|g_j\|_\infty + \|r_j\|_\infty \|o_j\|_\infty B_{U_c}\}$. We also have $\|h_t\|_2 \leq \sqrt{d}$. Thus, put together, we

have $\|h_t\|_2 \leq \min\left\{\sqrt{d}, B_{W_c}B_x\frac{\beta^t-1}{\beta-1}\right\}$. Next, we investigate the Lipschitz continuity of $h_t$.

$$\|h_t - h'_t\|_2 \leq \|o_t \odot \tanh(c_t) - o'_t \odot \tanh(c'_t)\|_2$$
$$\leq \|o_t - o'_t\|_2\|\tanh(c_t)\|_\infty + \|o'_t\|_\infty\|c_t - c'_t\|_2$$

We have to expand $o_t - o'_t$,

$$\|o_t - o'_t\|_2 \leq B_x\|W_o - W'_o\|_2 + B_{U_o}\|h_{t-1} - h'_{t-1}\|_2 + \|h_{t-1}\|_2\|U_o - U'_o\|_2.$$

Note that $\|B_{U_o}\|_2$ is usually small, $o_t$ and $o'_t$ are close, and we have $\|h_{t-1} - h'_{t-1}\|_2 \leq \|o_t\|_\infty\|c_{t-1} - c'_{t-1}\|_2 \leq \|c_{t-1} - c'_{t-1}\|_2$. Thus, we can derive

$$\|o_t - o'_t\|_2 \leq B_x\|W_o - W'_o\|_2 + B_{U_o}\|c_{t-1} - c_{t-1}\|_2 + \sqrt{d}\|U_o - U'_o\|_2.$$

We also expand $c_t - c'_t$ to get,

$$\|c_t - c'_t\|_2 \leq \|c_{t-1}\|_\infty\|g_t - g'_t\|_2 + \|r'_t\|_\infty\|c_{t-1} - c'_{t-1}\|_2 + \|\widetilde{c}_t\|_\infty\|r_t - r'_t\|_2 + \|r'_t\|_\infty\|\widetilde{c}_t - \widetilde{c}'_t\|_2.$$

We also have,

$$\|\widetilde{c}_t - \widetilde{c}'_t\|_2 \leq B_{U_c}\|h_{t-1} - h'_{t-1}\|_2 + \|h_{t-1}\|_2\|U_c - U'_c\|_2 + B_x\|W_c - W'_c\|_2,$$
$$\|g_t - g'_t\|_2 \leq B_x\|W_g - W'_g\|_2 + B_{W_g}\|h_{t-1} - h'_{t-1}\|_2 + \sqrt{d}\|U_g - U'_g\|_2, \quad \text{and}$$
$$\|r_t - r'_t\|_2 \leq B_x\|W_r - W'_r\|_2 + B_{W_r}\|h_{t-1} - h'_{t-1}\|_2 + \sqrt{d}\|U_r - U'_r\|_2.$$

Putting together, we get

$$\|c_t - c'_t\|_2$$
$$\leq B_x\left(\|W_c - W'_c\|_2 + \|W_g - W'_g\|_2 + \|W_r - W'_r\|_2\right)$$
$$\quad + \sqrt{d}\left(\|U_c - U'_c\|_2 + \|U_g - U'_g\|_2 + \|U_r - U'_r\|_2\right)$$
$$\quad + \|g_t\|_\infty\|c_{t-1} - c'_{t-1}\|_2 + \left(\|r_t\|_\infty B_{U_c} + B_{U_g} + B_{U_r}\right)\|h_{t-1} - h'_{t-1}\|_2$$
$$\leq B_x\left(\|W_c - W'_c\|_2 + \|W_g - W'_g\|_2 + \|W_r - W'_r\|_2 + (B_{U_c} + B_{U_g} + B_{U_r})\|W_o - W'_o\|_2\right)$$
$$\quad + \sqrt{d}\left(\|U_c - U'_c\|_2 + \|U_g - U'_g\|_2 + \|U_r - U'_r\|_2 + (B_{U_c} + B_{U_g} + B_{U_r})\|U_o - U'_o\|_2\right)$$
$$\quad + \left(\|o_t\|_\infty\|r_t\|_\infty B_{U_c} + B_{U_g} + B_{U_r} + B_{U_o}\right)\|c_{t-1} - c'_{t-1}\|_2.$$

By induction, we have

$$\|c_t - c'_t\|_2$$
$$\leq B_x\frac{\theta^t - 1}{\theta - 1}\left(\|W_c - W'_c\|_2 + \|W_g - W'_g\|_2 + \|W_r - W'_r\|_2 + (B_{U_c} + B_{U_g} + B_{U_r})\|W_o - W'_o\|_2\right)$$
$$\quad + \sqrt{d}\frac{\theta^t - 1}{\theta - 1}\left(\|U_c - U'_c\|_2 + \|U_g - U'_g\|_2 + \|U_r - U'_r\|_2 + (B_{U_c} + B_{U_g} + B_{U_r})\|U_o - U'_o\|_2\right),$$

where $\theta = \beta + B_{U_g} + B_{U_r} + B_{U_o}$. Now we immediately have

$$\|h_t - h'_t\|_2$$
$$\leq B_x\frac{\theta^t - 1}{\theta - 1}\left(\|W_c - W'_c\|_2 + \|W_g - W'_g\|_2 + \|W_r - W'_r\|_2 + (B_{U_c} + B_{U_g} + B_{U_r})\|W_o - W'_o\|_2\right)$$
$$\quad + \sqrt{d}\frac{\theta^t - 1}{\theta - 1}\left(\|U_c - U'_c\|_2 + \|U_g - U'_g\|_2 + \|U_r - U'_r\|_2 + (B_{U_c} + B_{U_g} + B_{U_r})\|U_o - U'_o\|_2\right).$$

Then the Lipschitz continuity of $y_t$ can be written as

$$\|y_t - y'_t\|_2 \leq \rho_y B_V\|h_t - h'_t\|_2 + \rho_y\sqrt{d}\|V - V'\|_2.$$

Following the same argument for proving the generalization bound of vanilla RNNs, we can get the generalization bound for LSTM RNNs as

$$\mathbb{P}\left(\widetilde{z}_t \neq z_t\right) \leq \widehat{\mathcal{R}}_\gamma(f_t) + O\left(\frac{d\rho_y B_V \min\left\{\sqrt{d}, B_{W_c}B_x\frac{\beta^t-1}{\beta-1}\right\}\sqrt{\log\left(d\sqrt{m}\frac{\theta^t-1}{\theta-1}\right)}}{\sqrt{m}\gamma} + \sqrt{\frac{\log\frac{1}{\delta}}{m}}\right).$$

$\square$

## D    EXTENSION TO CONVOLUTIONAL RNNS

We further extend our analysis to Convolutional RNNs (Conv RNNs). Conv RNNs integrate convolutional filters and recurrent neural networks. Specifically, we consider input $x \in \mathbb{R}^d$ and $k$-channel $k$-dimensional convolutional filters $\mathcal{I}_1, \ldots, \mathcal{I}_k \in \mathbb{R}^k$ followed by an average pooling layer over the $k$ channels for reducing dimensionality. Extensions to convolution with strides and other kinds of average pooling layers (e.g., blockwise pooling) are straightforward.

Here we denote the circulant-like matrix generated by $\mathcal{I}_i$ as

$$
C_i = \begin{bmatrix} \mathcal{I}_i^\top & \underbrace{0 \ldots \ldots \ldots 0}_{d-k} \\ 0 & \mathcal{I}_i^\top & \underbrace{0 \ldots \ldots 0}_{d-k-1} \\ \vdots & \ddots & \vdots \\ \underbrace{0 \ldots \ldots 0}_{d-k-1} & \mathcal{I}_i^\top & 0 \\ \underbrace{0 \ldots \ldots \ldots 0}_{d-k} & \mathcal{I}_i^\top \end{bmatrix} \in \mathbb{R}^{(d-k+1) \times d},
$$

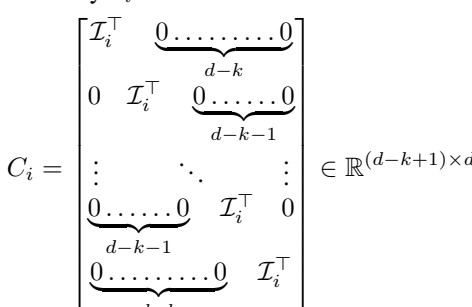
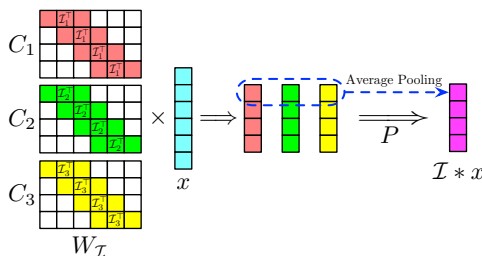

Figure 5: Illustration of input $x \in \mathbb{R}^6$ convolving with 3-channel 3-dimensional convolutional filters $\mathcal{I}_1, \mathcal{I}_2$, and $\mathcal{I}_3$, followed by an average pooling.

and write $W_{\mathcal{I}} = [C_1^\top, \ldots, C_k^\top]^\top$. We further denote $P = \frac{1}{k} \underbrace{[I_{d-k+1} \; I_{d-k+1} \; \cdots \; I_{d-k+1}]}_{\text{totally } k \text{ identity matrices}}$, where $I_d$ denotes the $d$-dimensional identity matrix. Define $\mathcal{I} = [\mathcal{I}_1, \ldots, \mathcal{I}_k]$, and $\mathcal{I} * x = P W_{\mathcal{I}} x$. Given a sample $(x_t, z_t)_{t=1}^T$, the Conv RNNs compute $h_t$ and $y_t$ as follows,

$$
h_t = \sigma_h \left( \mathcal{U} * h_{t-1} + \mathcal{W} * x_t \right), \quad \text{and} \quad y_t = \sigma_y \left( \mathcal{V} * h_t \right),
$$

where $h_t, x_t \in \mathbb{R}^d$, and $\mathcal{U}, \mathcal{V}, \mathcal{W} \in \mathbb{R}^{k \times k}$ are matrices with column vectors being $k$-dimensional convolutional filters. We use zero-padding to ensure the output dimension of convolutional filters matches the input (Krizhevsky et al., 2012). To get $y_t$, we convolve $h_t$ with $\mathcal{V}$ followed by an average pooling to reduce the dimension to $K$. Since we aim to show that Conv RNNs reduce the dependence on $d$ in generalization through parameter sharing, we simplify the notations to assume $h_0 = 0$, and impose the following assumption. Extensions to general settings are straightforward.

**Assumption 8.** The activation operators $\sigma_h$ and $\sigma_y$ are 1-Lipschitz with $\sigma_h(0) = \sigma_y(0) = 0$. $\sigma_h$ is entrywise bounded by 1. The convolutional filters $\mathcal{U}, \mathcal{V}$, and $\mathcal{W}$ are orthogonal with normalized columns, i.e., $\mathcal{U}^\top \mathcal{U} = \mathcal{U}\mathcal{U}^\top = \frac{1}{k} I_k, \mathcal{V}^\top \mathcal{V} = \mathcal{V}\mathcal{V}^\top = \frac{1}{k} I_k$, and $\mathcal{W}^\top \mathcal{W} = \mathcal{W}\mathcal{W}^\top = \frac{1}{k} I_k$.

We remark that the orthogonality constraints enhance the diversity among convolutional filters (Xie et al., 2017; Huang et al., 2017). Additionally, the normalization factor $\frac{1}{k}$ is to control the spectral norms of $W_{\mathcal{U}}, W_{\mathcal{V}}$, and $W_{\mathcal{W}}$, which prevents the blowup of hidden state. Denote by $\mathcal{F}_{c,t}$ the class of mappings from the first $t$ inputs to the $t$-th output computed by Conv RNNs. Then the generalization bound is given in the following theorem.

**Theorem 6.** Let activation operators $\sigma_h$ and $\sigma_y$ be given, and Assumptions 1 and 8 hold. Then for $(x_t, z_t)_{t=1}^T$ and $S = \left\{ (x_{i,t}, z_{i,t})_{t=1}^T, i = 1, \ldots, m \right\}$ drawn i.i.d. from any underlying distribution over $\mathbb{R}^{d \times T} \times \{1, \ldots, K\}$, with probability at least $1 - \delta$ over $S$, for every margin value $\gamma > 0$ and every $f_t \in \mathcal{F}_{c,t}$ for integer $t \le T$, we have

$$
\mathbb{P} \left( \widetilde{z}_t \neq z_t \right) \le \widehat{\mathcal{R}}_\gamma(f_t) + O \left( \frac{B_x k t \sqrt{\log \left( dt \sqrt{m} \right)}}{\sqrt{m} \gamma} + \sqrt{\frac{\log \frac{1}{\delta}}{m}} \right).
$$

The detailed proof is provided in D.1. Similar to the analysis of vanilla RNNs, our proof is based on the Lipschitz continuity of Conv RNNs with respect to its model parameters in the convolutional filters. Specifically, by Assumption 8, the spectral norms of $W_{\mathcal{U}}, W_{\mathcal{V}}$, and $W_{\mathcal{W}}$ are all bounded by 1. Combining with the inequality, $\|W_{\mathcal{U}}\|_F \le \sqrt{d}\|\mathcal{U}\|_F$, we have $\|y_t - y_t'\|_2 \le L_{V,t}\|\mathcal{V} - \mathcal{V}'\|_F + L_{\mathcal{U},t}\|\mathcal{U} - \mathcal{U}'\|_F + L_{\mathcal{W},t}\|\mathcal{W} - \mathcal{W}'\|_F$, where $L_{\mathcal{U},t}, L_{\mathcal{V},t}$, and $L_{\mathcal{W},t}$ are polynomials in $d$ and $t$. Additionally, observe that the total number of parameters in a Conv RNN is at most $3k^2$, which is independent of input dimension $d$. As a consequence, the generalization bound of Conv RNNs only has a lieanr dependence on $k$ and $t$.

### D.1 PROOF OF THEOREM 6

*Proof.* We first characterize the Lipschitz continuity of $\|y_t\|_2$ with respect to model parameters $\mathcal{U}$, $\mathcal{W}$ and $\mathcal{V}$. We have

$$\|y_t - y_t'\|_2 \le \rho_y \|h_t\|_2 \|W_\mathcal{V} - W_{\mathcal{V}'}\|_2 + \rho_y \|W_\mathcal{V}\|_2 \|h_t - h_t'\|_2.$$

Since $\|h_t\|_\infty \le 1$, we have $\|h_t\|_2 \le \sqrt{d}$. Then we expand $h_t - h_t'$,

$$
\begin{aligned}
\|h_t - h_t'\|_2 &\le \rho_h \|\mathcal{U}' * h_{t-1} + \mathcal{W} * x_t - \mathcal{U}' * h_{t-1}' - \mathcal{W}' * x_t\|_2 \\
&= \rho_h \|PW_\mathcal{U} h_{t-1} + PW_\mathcal{W} x_t - PW_\mathcal{U}' h_{t-1}' - PW_\mathcal{W} x_t\|_2 \\
&\le \rho_h \|P\|_2 \|W_\mathcal{U} h_{t-1} + W_\mathcal{W} x_t - W_{\mathcal{U}'} h_{t-1}' - W_{\mathcal{W}'} x_t\|_2 \\
&\le \rho_h \|P\|_2 \left[ B_x \|W_\mathcal{W} - W_{\mathcal{W}'}\|_2 + \sqrt{d}\|W_\mathcal{U} - W_{\mathcal{U}'}\|_2 + \|W_\mathcal{U}\|_2 \|h_{t-1} - h_{t-1}'\|_2 \right]
\end{aligned}
$$

Observe that we have by the definition of circulant matrix,

$$\|W_\mathcal{U} - W_{\mathcal{U}'}\|_2^2 \le \|W_\mathcal{U} - W_{\mathcal{U}'}\|_F^2 = (d-k)\|\mathcal{U} - \mathcal{U}'\|_F^2 \le d\|\mathcal{U} - \mathcal{U}'\|_F^2.$$

The same holds for $W_\mathcal{W} - W_{\mathcal{W}'}$ and $W_\mathcal{V} - W_{\mathcal{V}'}$. We also have $\|P\|_2 = 1$. The remaining task is to bound the spectral norm of $W_\mathcal{U}$ and $W_\mathcal{V}$. Consider the matrix product $W_\mathcal{U}^\top W_\mathcal{U}$. We claim that the diagonal elements of $W_\mathcal{U}^\top W_\mathcal{U}$ is bounded by $\sum_{i=1}^k \|\mathcal{U}_i\|_2^2$, and the off-diagonal elements are zero. To see this, denote by $C_{\mathcal{U}_i}$ the circulant like matrix generated by $\mathcal{U}_i$. Then we have $W_\mathcal{U} = [C_{\mathcal{U}_1}^\top, \ldots, C_{\mathcal{U}_k}^\top]^\top$. The diagonal elements of $W_\mathcal{U}^\top W_\mathcal{U}$ are

$$\left(W_\mathcal{U}^\top W_\mathcal{U}\right)_{ii} = \sum_{j=1}^k \left(C_{\mathcal{U}_j}^\top C_{\mathcal{U}_j}\right)_{ii} \le \sum_{i=1}^k \|\mathcal{U}_i\|_2^2.$$

By the orthogonality of $\mathcal{U}$, the off-diagonal elements are

$$\left(W_\mathcal{U}^\top W_\mathcal{U}\right)_{pq} = \sum_{j=1}^k \left(C_{\mathcal{U}_j}^\top C_{\mathcal{U}_j}\right)_{pq} = \sum_{j=1}^k \left(C_{\mathcal{U}_j}\right)_{:p}^\top \left(C_{\mathcal{U}_j}\right)_{:q} = 0.$$

Thus, the spectral norm $\|W_\mathcal{U}\|_2 \le \sqrt{\sum_{i=1}^k \|\mathcal{U}_i\|_2^2} \le 1$, and $\|W_\mathcal{V}\|_2, \|W_\mathcal{W}\|_2 \le 1$ also hold. Then we can derive

$$\|h_t - h_t'\|_2 \le \rho_h B_x \sqrt{d}\|\mathcal{W} - \mathcal{W}'\|_F + \rho_h d\|\mathcal{U} - \mathcal{U}'\|_F + \rho_h \|h_{t-1} - h_{t-1}'\|_2.$$

Apply the above inequality recursively, we get

$$
\begin{aligned}
\|h_t - h_t'\|_2 &\le \rho_h B_x \sqrt{d} \frac{\rho_h^t - 1}{\rho_h - 1}\|\mathcal{W} - \mathcal{W}'\|_F + \rho_h d \frac{\rho_h^t - 1}{\rho_h - 1}\|\mathcal{U} - \mathcal{U}'\|_F \\
&\le B_x \sqrt{d} t\|\mathcal{W} - \mathcal{W}'\|_F + dt\|\mathcal{U} - \mathcal{U}'\|_F.
\end{aligned}
$$

Thus, we have the following Lipschitz continuity of $\|y_t\|_2$,

$$\|y_t - y_t'\|_2 \le d\|\mathcal{V} - \mathcal{V}'\|_F + B_x \sqrt{d} t\|\mathcal{W} - \mathcal{W}'\|_F + dt\|\mathcal{U} - \mathcal{U}'\|_F.$$

We also bound the norm of $h_t$ by induction. Specifically, we have

$$\|h_t\|_2 \le \rho_h \|PW_\mathcal{U} h_{t-1} + PW_\mathcal{W} x_t\|_2 \le \rho_h \|W_\mathcal{U} h_{t-1}\|_2 + \rho_h \|W_\mathcal{W} x_t\|_2 \le \|h_{t-1}\|_2 + B_x.$$

Applying the above expression recursively, we have $\|h_t\|_2 \le \min\{\sqrt{d}, B_x t\} \le B_x t$. Then following the same argument for proving the generalization bound of vanilla RNNs, we can get the generalization bound for Conv RNNs as

$$\mathbb{P}\left(\widetilde{z}_t \ne z_t\right) \le \widehat{\mathcal{R}}_\gamma(f_t) + O\left(\frac{B_x k t \sqrt{\log\left(dt\sqrt{m}\right)}}{\sqrt{m}\gamma} + \sqrt{\frac{\log \frac{1}{\delta}}{m}}\right).$$

$\square$

