# OpenReview forum: "On Generalization Bounds of a Family of Recurrent Neural Networks"
_ICLR.cc/2019/Conference_

### Official Review · AnonReviewer1 · 2018-10-30
**limited technical novelty**

**Rating:** 3
**Confidence:** 4

**Review:**


The authors provide new generalization bounds for recurrent neural networks.
Their main result is a new bound for vanilla RNNs, but they also have
bounds for gated RNNs.

They claim that their vanilla bound improves on an earlier
bound for RNNs in Section 6 of an ICML'18 paper by Zhang, et al.
The main result of the submission is incomparable in strength with the earlier result,
because this submission assumes that the activation functions in the hidden
layers are bounded, where the earlier paper did not.  Part of the difference in the results
(roughly speaking, the "min" in the bound) can be traced to this difference in the assumptions.
 (This paper uses this assumption in the second-to-last line of the proof of Lemma 6.)

I think that the root cause of the remaining difference is that this paper,
at its core, adapts the more traditional analysis, used in Haussler's
1992 InfComp paper.  New analyses, like from the Bartlett, et al
NIPS'17 paper, strove for a weak dependence in the number of parameters,
but this proof technique appears to lead to a worse dependence on the
depth.  I think that, if you unwind the network, to view the function
from the first t positions of the input to output number t as a
depth t network, and apply Haussler's bound, you will get a qualitatively
similar result (in particular with bounds that scale polynomially with
d and t).  I think that Haussler's proof technique can be adapted to
take advantage of the weight sharing between layers in the unrolled
network.

It is somewhat interesting to note that the traditional bounds have
a better dependence on depth, with correspondingly better dependence
on the length of the output sequence of the RNN.

I also do not see that substantial new insight is gained through the
analysis that incorporates gating.

I do not see much technical novelty in this paper.

---

> ### Author Response · Authors · 2018-11-26
> **Response to Reviewer 1**
>
> 1. Comment:
> Compared to [1], it appears to me that the bounds are incomparable in strength, that the new bound has an improved dependence on the size of the output sequence, but a worse dependence on the number of hidden nodes.
>
> 1. Response:
> In the introduction on page 3, we have clearly indicated that we UNIFORMLY improve the generalization bound in [1] under ALL the three cases:
>
> Our complexity bound involves a minimization over \sqrt{d} and an exponential factor (||U||^t - 1) / (||U|| -1).
>
> When ||U|| < 1, our bound is tighter by a factor of t^2; When ||U|| = 1, our bound is tighter by a factor of t.
>
> Considering ||U|| > 1, if we take the minimization term as (||U||^t - 1) / (||U|| -1), our bound is already improves a factor of t^{3/2}. Note that, when t is large, \sqrt{d}  <<  (||U||^t - 1) / (||U|| -1), the minimization term is dominated by \sqrt{d}. Our bound is of the order \tilde{O}(\sqrt{d^3t} / \sqrt{m}\gamma), and is polynomial in d and t. We replace an extremely large exponential dependence on t by \sqrt{d}. As can be seen, we improve dependence on the size of the output sequence while not introducing extra factors on the number of hidden nodes.
>
> 2. Comment:
> I think that the root cause of this difference is that this paper, at its core, adapts the more traditional analysis, used in [2].
>
> It is somewhat interesting to note that the traditional bounds have a better dependence on depth, with correspondingly better dependence on the length of the output sequence of the RNN.
>
> 2. Response:
> Our analysis is quite different from [2]. In detail, Theorem 11 in [2] gives a covering number upper bound on mappings defined by feedforward networks. It adopts a layer wise evaluation based on the Lipschitz continuity of outputs with respect to inputs. Our analysis, however, characterizes the Lipschitz continuity of outputs with respect to weight matrices. This new idea allows us to avoid complicated layer wise evaluation, and decouple the spectral norms of weight matrices and the number of parameters (above Figure 1 on page 2).
>
> Moreover, our generalization bound improves the dependence on depth compared to [2]. Specifically, the bound in Theorem 11 in [2] involves a product of Lipschitz constants of each layer up to the length of the input sequence, and assumes the Lipschitz constant greater than 1. Hence, the resulting bound is valid in limited scenarios and inevitably exponential in t. Our bound actually is always polynomial in d and t as discussed in the introduction on page 3.
>
> 3. Comment:
> I also do not see that substantial new insight is gained through the analysis that incorporates gating.
>
> 3. Response:
> Our contribution is providing rigorous theoretical justifications on the advantages of MGU and LSTM RNNs in generalization, which goes beyond the vague intuition of introducing gated signals.
>
> We establish generalization bounds for MGU and LSTM RNNs in Section 5. To our best knowledge, it is the first time that such generalization bounds are established for these RNNs. We further demonstrate their advantages over vanilla RNNs in generalization based on the results of Theorems 4 and 5 via concrete examples (under Theorems 4 and 5). It turns out that both gated RNNs introduce extra decaying factors to further reduce the dependence on d and t in generalization.
>
> 4. Comment:
> I do not see much technical novelty in this paper.
>
> 4. Response:
> We provide new understandings of RNNs by connecting their generalization properties to their empirical success. We establish generalization bounds using the PAC-learning framework by instilling our simple but new complexity analysis for RNNs. In particular, we characterize the Lipschitz property of the output of RNNs with respect to weight matrices (Lemma 2). This key step allows us to construct a covering on RNNs from simple coverings of weight matrices, which results in a complexity bound of RNNs polynomial in d and t.. We further extend our analysis to establish the first generalization bounds for MGU and LSTM RNNs.
>
> To sum up, the main contributions are listed as follows (see paragraphs 2 - 5 on page 3):
> 1) We develop new techniques for characterizing the complexity of RNNs, which allows us to establish tight generalization bounds. We justify that the complexity of RNNs does not suffer from significant curse of dimensionality.
> 2) We establish generalization bounds for MGU and LSTM RNNs, and justify their advantages over vanilla RNNs (under Theorems 4 and 5) by introducing gated signals.
>
> Given very limited literature on RNNs and their variants (not even mentioning that most of existing theoretical results are negative, e.g., exponential complexities), we see no reason why our paper is not novel.
>
> Reference
> [1] Zhang, Jiong, et al. "Stabilizing Gradients for Deep Neural Networks via Efficient SVD Parameterization." 2018.
> [2] Haussler, David. "Decision theoretic generalizations of the PAC model for neural net and other learning applications." 1992.

---

> > ### Comment · AnonReviewer1 · 2018-11-27
> > **have read response and updated review**
> >
> > I have read the authors' response, given this submission and some preceding papers a closer look, and updated my review.
> >
> > I still believe that Part 2 of Corollary 3 of [2], given
> > reasonably-sized weights and a squashing function with a Lipschitz
> > constant of 1, such as tanh or the ReLU, provides a sample complexity
> > bound polynomial in the number of weights and the depth.  As seen in
> > the statement of that corollary and elsewhere in the paper, the
> > requirement is only that the Lipschitz constant is at least 1, not
> > greater than 1.  Note that Theorem 11 of [2] bounds a covering number, and that,
> > as the authors no doubt know, sample complexity bounds are typically
> > logarithmic in covering-number bounds.  Since, in the worst case,
> > the Lipschitz constant of a layer of a fully connected network
> > is the product of the operator norm of the layer's weight matrix
> > and the Lipschitz constant of the activation function, Lemma 3 of
> > the current paper looks similar to me to Theorem 11 of [2].

---

> > > ### Author Response · Authors · 2018-11-29
> > > **Comparison to [1] and difference from Theorem 11 of [2]**
> > >
> > > Our bound does improve the bound in [1] even if we drop the boundedness assumption. As you have noticed, the boundedness assumption (Assumption 3) only contributes a \sqrt{d} term in the “min” part (and we have pointed out that [1] fail to consider the boundedness of hidden states in the introduction). If we drop the boundedness assumption, our complexity bound is \tilde{O}(d (||U||^t -1)/(||U|| - 1) \sqrt{\log (||U||^t -1)/(||U|| - 1)}). Compared to [1]:
> > > When ||U|| < 1, we improve a t^2 factor;
> > > When ||U|| = 1, we improve a t factor;
> > > When ||U|| > 1, we improve a t^{3/2} factor.
> > > Therefore, our bound still uniformly improves the bound in [1]. Moreover, the bound in [1] only considers the ReLU activation (Theorem 5 in [1]), while our result holds for general Lipschitz activations.
> > >
> > > Next, as mentioned in the introduction, [2] adopts a layer wise analysis and builds upon the Lipschitz continuity of each layer with respect to the INPUT in a feedforward neural network. [2] then constructs a covering for the whole network by induction. Our analysis is quite different, which characterizes the overall Lipschitz continuity of RNNs with respect to WEIGHT MATRICES.
> > >
> > > More importantly, there is a huge difference between characterizing the Lipschitz continuity of feedforward neural networks and RNNs. Note that, RNNs recurrently take input, and each layer uses the previous hidden state and a NEW DATA INPUT to compute the next hidden state and the output. In feedforward neural networks, there is literally only ONE input to the whole network. Therefore, there is no easy way to evaluate the Lipschitz continuity of each layer in RNNs, considering such a Lipschitz continuity with respect to input may even not be well defined.
> > >
> > > Lastly, Theorem 11 in [2] and Lemma 3 of our paper are actually very different, though they are both based on volume ratio argument for coverings. The exponent of Theorem 11 in [2] is the total number of weights in feedforward neural networks, which is d^2 t. The exponent in our Lemma 3 is only d^2, which meets the total number of parameters in RNNs. Handling parameter sharing is nontrivial given the recurrent nature and the sequence input of RNNs (e.g., the Lipschitz continuity of each layer in RNNs is already ambiguous). Our analysis then makes it clear by exploiting the parametric form of RNNs and characterizing their Lipschitz continuity with respect to weight matrices.
> > >
> > > Reference
> > > [1] Zhang, Jiong, et al. "Stabilizing Gradients for Deep Neural Networks via Efficient SVD Parameterization." 2018.
> > > [2] Haussler, David. "Decision theoretic generalizations of the PAC model for neural net and other learning applications." 1992.

---

### Official Review · AnonReviewer2 · 2018-11-02
**The paper focuses on the generalization performance of RNNs and its variant in a theoretical perspective.**

**Rating:** 6
**Confidence:** 4

**Review:**

The paper focuses on the generalization performance of RNNs and its variant in a theoretical perspective. Compared to the previous result (Zhang et al., 2018) for RNNs, this paper refines the generalization bounds for vanilla RNNs in all cases and fills the blank for RNN variants like MGU and LSTM. Specifically, in the work of Zhang et al. (2018), the complexity term quadratically depends on the layer (or say, current sequence length, denoted by t in original paper), making it less instructive. This paper improves it to at most linear dependence and can achieve at logarithmic dependence in some cases, which should be accredited.

The key step in the proof is Lemma 2. In Lemma 2, the spectral norms of weight matrices and the number of weight parameters are decoupled. With Lemma 2, it is natural to construct a epsilon-net for RNNs and then upper bound the empirical Rademacher complexity by Dudley’s entropy integral, since such methodology is not so novel. Bartlett, et al. (2017) developed this technique to analyze the generalization bound for neural networks in a margin-based multiclass classification. However, it seems a little unexplainable to apply a technique developed from classification to analyze RNNs, since the main task of RNNs never should be classification. I wonder the motivation of analyzing generalization of RNNs by the techniques established by Bartlett.

---

> ### Author Response · Authors · 2018-11-26
> **Response to Reviewer 2**
>
> 1. Comment:
> However, it seems a little unexplainable to apply a technique developed from classification to analyze RNNs, since the main task of RNNs never should be classification.
>
> 1. Response:
> We study seq2seq classification tasks since they have been widely used in real world applications for RNNs. To name a few, in speech recognition, [1] hybridizes hidden Markov model with RNNs to label unsegmented sequence data; In computer vision, [2, 3] demonstrate scene labeling with LSTM and RNNs, achieving higher accuracy than baseline methods; In healthcare, [4] proposes a model, Doctor AI, to perform multiple label prediction (one for each disease or medication category). In addition, [5, 6] both apply RNNs to real-world healthcare datasets (MIMIC-III, PhysioNet, and ICU data) for mortality prediction and other multiple classifications tasks. We establish bounds for classification because it is typical in learning theory and is easy to compare among existing literature.
>
> On the other hand, our analysis applies in other tasks as long as a suitable Lipschitz loss function is chosen. Specifically, Lemma 4 establishes an upper bound for empirical Rademacher complexity of general Lipschitz loss functions (the last line in Appendix A.4). By replacing the loss function in Lemma 1, we can derive generalization bounds for various tasks other than classification.
>
> References
> [1] Graves, Alex, Santiago Fernández, Faustino Gomez, and Jürgen Schmidhuber. "Connectionist temporal classification: labelling unsegmented sequence data with recurrent neural networks." In Proceedings of the 23rd international conference on Machine learning, pp. 369-376. ACM, 2006.
> [2] Byeon, Wonmin, Thomas M. Breuel, Federico Raue, and Marcus Liwicki. "Scene labeling with lstm recurrent neural networks." In Proceedings of the IEEE Conference on Computer Vision and Pattern Recognition, pp. 3547-3555. 2015.
> [3] Socher, Richard, Cliff C. Lin, Chris Manning, and Andrew Y. Ng. "Parsing natural scenes and natural language with recursive neural networks." In Proceedings of the 28th international conference on machine learning (ICML-11), pp. 129-136. 2011.
> [4] Choi, Edward, Mohammad Taha Bahadori, Andy Schuetz, Walter F. Stewart, and Jimeng Sun. "Doctor ai: Predicting clinical events via recurrent neural networks." In Machine Learning for Healthcare Conference, pp. 301-318. 2016.
> [5] Che, Zhengping, Sanjay Purushotham, Kyunghyun Cho, David Sontag, and Yan Liu. "Recurrent neural networks for multivariate time series with missing values." Scientific reports 8, no. 1 (2018): 6085.
> [6] Xu, Yanbo, Siddharth Biswal, Shriprasad R. Deshpande, Kevin O. Maher, and Jimeng Sun. "RAIM: Recurrent Attentive and Intensive Model of Multimodal Patient Monitoring Data." In Proceedings of the 24th ACM SIGKDD International Conference on Knowledge Discovery & Data Mining, pp. 2565-2573. ACM, 2018.

---

### Official Review · AnonReviewer3 · 2018-11-02
**Marginal improvement over existing bounds; Incomplete comparison with existing works**

**Rating:** 4
**Confidence:** 4

**Review:**

This paper studies the generalization properties of Recurrent Neural Networks (RNN) and its variants for the sequence to sequence multiclass prediction problem. The problem is important to understand in the theoretical machine learning community. The paper is written well overall, clearly explaining the results obtained. I would like to raise several important points:

1. Missing comparison with parameter counting bounds: there has been a long line of research on generalization bounds for RNNs by obtaining bounds on the VC dimension of the function class [1, 2] which provide generalization bounds for various non-linearities. The bounds obtained are polynomial in the sequence length T (often sublinear or linear) and should be compared with the existing bounds for a thorough comparison.

2. Vacuous bounds in the regime \beta >1: Most recurrent architectures with no restrictions on the transition matrices work in the regime where they are more expressive and \beta >1. A quick glance at Table 1 suggests that the bounds obtained through Theorem 3 are exponential in t and are mostly vacuous. They can indeed be subsumed by generalization bounds based on VC theory. The bound obtained in Theorem 2 comes rather easily from the bounded assumption on the non-linearity and is this not very interesting.

3. Technical contribution: While the authors propose the first bounds for LSTMs and MGUs, most of the analysis seems to be a marginal contribution over the work of Bartlett et al. [3]

4. Missing experiments to validate nature of bounds: Bartlett et al. [3] performed extensive experiments to exhibit the correct scaling of the generalization bounds with the "model complexity" introduced upto numerical constants. It would be good to have some experiments in the sequence to sequence setting to understand if the obtained complexities are in fact what one would expect in practice.


[1] Koiran, Pascal, and Eduardo D. Sontag. "Vapnik-Chervonenkis Dimension of Recurrent Neural Networks." Discrete Applied Mathematics 86.1 (1998): 63-79.
[2] Dasgupta, Bhaskar, and Eduardo D. Sontag. "Sample complexity for learning recurrent perceptron mappings." Advances in Neural Information Processing Systems. 1996.
[3] Bartlett, Peter L., Dylan J. Foster, and Matus J. Telgarsky. "Spectrally-normalized margin bounds for neural networks." Advances in Neural Information Processing Systems. 2017.

---

> ### Author Response · Authors · 2018-11-26
> **Response to Reviewer 3**
>
> 1. Comment:
> Missing comparison with parameter counting bounds [1, 2].
>
> 1. Response:
> [1, 2] are early works on RNNs and their analysis is based on very simple network architectures, which is not directly comparable to our results. Specifically, [1] only consider RNNs taking the first entry of the hidden state as output and restrict their discussion to polynomial or sigmoid activation functions. Moreover, their analysis is based on unwinding RNNs as feedforward neural networks and adopt a layer-wise analysis, which fails to incorporate the parameter sharing in RNNs, and the established bounds are far from satisfactory (O(d^8 t^2) for sigmoid activation). [2] simply focus on linear RNNs for binary classification problems, its extension to general settings are quite unclear.
>
> 2. Comment:
> Vacuous bounds in the regime \beta >1.
>
> 2. Response:
> We have clearly indicated that our bound is always polynomial in d and t in the introduction on page 3, which is not vacuous.
>
> Moreover, both bounds of Theorem 3 are obtained under the same assumptions as in Theorem 2 with additional norm constraints on weight matrices. The exponential term stems from the layer wise covering argument rather than the range of the output. The bound in Theorem 2 is still polynomial in d and t, since we exploit the parametric form of RNNs and construct the covering by weight matrix coverings.
>
> Existing literature has shown that keeping the spectral norm of weight matrix U close to 1 stabilizes the training of RNNs. This can be achieved by orthogonal initialization and imposing extra constraints or regularization [3-5]. We further discuss the trade-off between representation and generalization beneath Theorem 2 on page 4: \beta \approx 1 helps balance the generalization and representation of RNNs.
>
> 3. Comment:
> Technical contribution: marginal.
>
> 3. Response:
> We provide new understandings of RNNs by connecting their generalization properties to their empirical success. We establish generalization bounds using the PAC-learning framework by instilling our simple but new complexity analysis for RNNs. In particular, we characterize the Lipschitz property of the output of RNNs with respect to weight matrices (Lemma 2). This key step allows us to construct a covering on RNNs from simple coverings of weight matrices, which results in a complexity bound of RNNs polynomial in d and t. We further extend our analysis to establish the first generalization bounds for MGU and LSTM RNNs.
>
> The proposed analysis is quite different from [6], which adopts a layer-wise evaluation for neural networks and constructs the overall covering via a complicated matrix covering argument. Moreover, as compared in Section 4 Table 1, our generalization bound is much tighter when \beta > 1.
>
> To sum up, the main contributions are listed as follows (see paragraphs 2 - 5 on page 3):
> 1) We develop new techniques for effectively characterizing the model class of RNNs, which allows us to establish tight generalization bounds.
> 2) We establish generalization bounds for both MGU and LSTM RNNs, and demonstrate their advantages over vanilla RNNs in generalization (under Theorems 4 and 5).
>
> Given very limited literature on RNNs and their variants (not even mentioning that most of existing theoretical results are negative, e.g., exponential complexities), we believe this paper has its novel contributions.
>
> 4. Comment:
> Missing experiments to validate nature of bounds.
>
> 4. Response:
> Please refer to the revised version for numerical evaluations in Section 6. In particular, we illustrate that our obtained generalization bound (Theorem 2) is much smaller than existing bounds even for \beta > 1.
>
> References
> [1] Koiran, Pascal, and Eduardo D. Sontag. "Vapnik-Chervonenkis Dimension of Recurrent Neural Networks." Discrete Applied Mathematics 86, no. 1 (1998): 63-79.
> [2] Dasgupta, Bhaskar, and Eduardo D. Sontag. "Sample complexity for learning recurrent perceptron mappings." In Advances in Neural Information Processing Systems, pp. 204-210. 1996.
> [3] Vorontsov, Eugene, Chiheb Trabelsi, Samuel Kadoury, and Chris Pal. "On orthogonality and learning recurrent networks with long term dependencies." arXiv preprint arXiv:1702.00071 (2017).
> [4] Arjovsky, Martin, Amar Shah, and Yoshua Bengio. "Unitary evolution recurrent neural networks." In International Conference on Machine Learning, pp. 1120-1128. 2016.
> [5] Simonyan, Karen, and Andrew Zisserman. "Very deep convolutional networks for large-scale image recognition." arXiv preprint arXiv:1409.1556 (2014).
> [6] Bartlett, Peter L., Dylan J. Foster, and Matus J. Telgarsky. "Spectrally-normalized margin bounds for neural networks." In Advances in Neural Information Processing Systems, pp. 6240-6249. 2017.

---

> > ### Comment · AnonReviewer3 · 2018-11-30
> > **Response to rebuttal**
> >
> > I have read the response provided by the authors , the reviews of the other authors and have also looked at the updated version of the paper for the changes made.
> >
> > The authors have added in the numerical numbers and comparison with previous bounds. However, as has also been pointed by Reviewer 1, the paper seems to have little technical novelty over existing methods. I would again like to reiterate that in the interesting regime \beta > 1,  this paper gets around the exponential bound by using a boundedness argument for the non-linearity (and the remaining analyses remains very similar to previous work).

---

### Public Comment · (anonymous) · 2018-10-05
**i.i.d data for RNN? Very Strong and Unrealistic Assumptions.**

I have quickly gone through this paper and find that the main part (including many technical details) of this paper is strongly based on the work of Peter Bartlett [1], which derives a margin bound for deep neural networks in the supervised multi-class classification setting. However, as we all know, when we derive the margin bound for supervised learning, we assume that the data are drawn i.i.d from some unknown distributions and therefore the generalization bound can be upper bounded by Rademacher Complexities plus a confidence term.

However, for RNNs, the i.i.d. assumption (also for block-by-block i.i.d) is not the case, as the training data arrive sequentially and they have correlations to each other. Therefore, we cannot assume that the training data are i.i.d. and all results in supervised learning cannot be  directly applied to analyze the generalization property for RNNs. There are indeed some works that try to analyze the generalization property when learning with sequential data, such as sequential Rademacher Complexities [2]. I strongly suggest the author(s) expose sufficient related works in this paper.

Furthermore, perhaps more important, the analysis of this paper is quite incremental, by adapting the work in [1] from DNNs to RNNs. The technical contributions are really not enough to our machine learning theory community.



References:

[1] Bartlett, Peter L., Dylan J. Foster, and Matus J. Telgarsky. "Spectrally-normalized margin bounds for neural networks." Advances in Neural Information Processing Systems. 2017.

[2] Rakhlin, Alexander, Karthik Sridharan, and Ambuj Tewari. "Online learning via sequential complexities." The Journal of Machine Learning Research 16.1 (2015): 155-186.

---

> ### Author Response · Authors · 2018-10-09
> **On i.i.d. data assumption and contributions**
>
> We appreciate your interest in our paper. It seems, however, that you are missing some key points in the paper.
>
> 1. Comment:
> For RNNs, the i.i.d. assumption (also for block-by-block i.i.d) is not the case.
>
> 1. Response:
> We consider i.i.d. input sequences (next to Figure 1 on page 2), not i.i.d. data. This allows data dependency within a sequence. Moreover, assuming i.i.d. sequences is reasonable in many seq2seq tasks. In image captioning, an image and its corresponding descriptions are naturally independent across pairs [1]. In health care, RNNs can be used to predict the physician diagnosis and medication order of the next visit [2]. Each patient displays independent symptoms across time, though, these symptoms are correlated for an individual.
>
> From a theoretical perspective, if we further assume i.i.d. data input, the Rademacher complexity can be reduced to O(complexity/ \sqrt{mT}). Compared to the presented bound O(complexity/ \sqrt{m}) (Theorem 1, equation (1)), we lose the extra \sqrt{T} factor due to data dependency.
>
> 2. Comment:
> There are indeed some works that try to analyze the generalization property when learning with sequential data, such as sequential Rademacher Complexities [3].
>
> 2. Response:
> To extend the analysis to full dependent data is quite challenging and still largely open. [3] derives complexity bounds for neural networks in Proposition 15. The result, however, is nearly negative. Since even with spectral normalization, the complexity bound is still exponential in the depth. Moreover, sequential Rademacher Complexity relies on the binary tree characterization of the labels, which restricts its application to binary classification problems. [4] also derives generalization bounds for dependent data, but under strong assumptions. Specifically, the sample sequence is \beta-mixing and they assume block independence after a sub-sample selection trick (Section 2 in [4]). With these difficulties, to extend our theory to fully dependent sequences is beyond the scope of the paper, and we leave it for future investigation.
>
> 3. Comment:
> Furthermore, perhaps more important, the analysis of this paper is quite incremental, by adapting the work in [5] from DNNs to RNNs. The technical contributions are really not enough to our machine learning theory community.
>
> 3. Response:
> Our work is based on the PAC-learning framework, which is widely adapted by most of the learning theory papers, including [5-8] for neural networks. The key difference among these papers is how to characterize the model class complexities.
>
> We propose a simple and effective way to evaluate RNNs, which is quite different from [5]. As discussed in the introduction, our generalization bound (Theorem 2) is tighter than [8], which follows the idea of [5] and utilizes the technique in [7]. Our analysis, however, decouples the spectral norms of weight matrices and the number of parameters by exploiting the parametric form of RNNs (above Figure 1 on page 2). In particular, we characterize the Lipschitz property of the output of RNNs with respect to weight matrices (Lemma 2). This key step allows us to construct a covering on RNNs from simple coverings of weight matrices. Note that, [5] adopts a layer-wise evaluation for neural networks and constructs the overall covering via a complicated matrix covering argument. Moreover, as compared in Section 4 Table 1, our generalization bound is much tighter when \beta > 1 and is still better for \beta \leq 1, only except when the weight matrices have low stable rank, and the input sequences are short.
>
> To sum up, the main contributions are listed as follows (see paragraphs 2 - 5 on page 3):
> 1) We develop new techniques for effectively characterizing the model class of RNNs, which allows us to establish tight generalization bounds. We also present refined bounds as a complementarity when additional norm constraints are available and compare among them.
> 2) We establish generalization bounds for both MGU and LSTM RNNs, and demonstrate their advantages over vanilla RNNs in generalization (under Theorems 4 and 5).
>
> References
> [1] Karpathy, Andrej, et al. "Deep visual-semantic alignments for generating image descriptions." 2015.
> [2] Choi, Edward, et al. "Doctor ai: Predicting clinical events via recurrent neural networks." 2016.
> [3] Rakhlin, Alexander, et al. "Online learning via sequential complexities." 2015.
> [4] Kuznetsov, Vitaly, et al. "Generalization bounds for non-stationary mixing processes." 2017.
> [5] Bartlett, Peter L., et al. "Spectrally-normalized margin bounds for neural networks." 2017.
> [6] Golowich, Noah, et al. "Size-Independent Sample Complexity of Neural Networks." 2017.
> [7] Neyshabur, Behnam, et al. "A pac-bayesian approach to spectrally-normalized margin bounds for neural networks." 2017.
> [8] Zhang, Jiong, et al. "Stabilizing Gradients for Deep Neural Networks via Efficient SVD Parameterization." 2018.

---

### Public Comment · (anonymous) · 2018-10-05
**It's a density estimation not classification problem**

In real-world applications, RNNs are applied to sequential data analysis for a density estimation problem, but not a classification problem. This paper aims at a wrong target.

---

> ### Author Response · Authors · 2018-10-09
> **Classification problem and its generalization bound**
>
> We study seq2seq classification tasks since they have been widely used in real world applications. To name a few, in speech recognition, [1] hybridizes hidden Markov model with RNNs to label unsegmented sequence data; In computer vision, [2, 3] demonstrate scene labeling with LSTM and RNNs, achieving higher accuracy than baseline methods; In healthcare, [4] proposes a model, Doctor AI, to perform multiple label prediction (one for each disease or medication category). In addition, [5, 6] both apply RNNs to real-world healthcare datasets (MIMIC-III, PhysioNet, and ICU data) for mortality prediction and other multiple classifications tasks. We establish bounds for classification because it is typical in learning theory and is easy to compare with the existing literature.
>
> On the other hand, for density estimation, our analysis applies as long as a suitable Lipschitz loss function is chosen (e.g. softmax with entropy loss). Specifically, Lemma 4 establishes an upper bound for empirical Rademacher complexity of general Lipschitz loss functions (the last line in Appendix A.4). By replacing the loss function in Lemma 1, we can derive generalization bounds for density estimation tasks, which can be covered as a regression problem. The bound is nearly identical to the established classification bound. The only difference is that we apply regression generalization bound and include the output range (see Chapters 3 and 10 in [7]). We will include a corresponding discussion in the next version.
>
> References
> [1] Graves, Alex, Santiago Fernández, Faustino Gomez, and Jürgen Schmidhuber. "Connectionist temporal classification: labelling unsegmented sequence data with recurrent neural networks." In Proceedings of the 23rd international conference on Machine learning, pp. 369-376. ACM, 2006.
> [2] Byeon, Wonmin, Thomas M. Breuel, Federico Raue, and Marcus Liwicki. "Scene labeling with lstm recurrent neural networks." In Proceedings of the IEEE Conference on Computer Vision and Pattern Recognition, pp. 3547-3555. 2015.
> [3] Socher, Richard, Cliff C. Lin, Chris Manning, and Andrew Y. Ng. "Parsing natural scenes and natural language with recursive neural networks." In Proceedings of the 28th international conference on machine learning (ICML-11), pp. 129-136. 2011.
> [4] Choi, Edward, Mohammad Taha Bahadori, Andy Schuetz, Walter F. Stewart, and Jimeng Sun. "Doctor ai: Predicting clinical events via recurrent neural networks." In Machine Learning for Healthcare Conference, pp. 301-318. 2016.
> [5] Che, Zhengping, Sanjay Purushotham, Kyunghyun Cho, David Sontag, and Yan Liu. "Recurrent neural networks for multivariate time series with missing values." Scientific reports 8, no. 1 (2018): 6085.
> [6] Xu, Yanbo, Siddharth Biswal, Shriprasad R. Deshpande, Kevin O. Maher, and Jimeng Sun. "RAIM: Recurrent Attentive and Intensive Model of Multimodal Patient Monitoring Data." In Proceedings of the 24th ACM SIGKDD International Conference on Knowledge Discovery & Data Mining, pp. 2565-2573. ACM, 2018.
> [7] Mohri, Mehryar, Afshin Rostamizadeh, and Ameet Talwalkar. Foundations of machine learning. MIT press, 2012.

---

### Author Response · Authors · 2018-11-26
**Updates in the revised version**

Please see our revised version, where we made the following changes:

1) We add a detailed discussion of existing literature on complexity bounds of neural networks in the introduction on page 2;

2) We provide numerical evaluations in Section 6 by comparing our generalization bound with existing ones.

---

### Meta-Review · Area_Chair1 · 2018-12-16
**Rewrite needed**

**Confidence:** 3
**Recommendation:** Reject

**Metareview:**

Some expert reviewers have raised novelty issues, that the authors have addressed in detail. Still, these expert reviewers are not entirely convinced. If this were a journal, I would recommend a major revision or reject-and-resubmit in order to allow the authors to anticipate the reviewers' concerns in the body of the paper and get some fresh reviews. I compliment the authors on the diligence they have put into the rebuttal stage, and look forward to reading the next version of the work. I will note that the bounds by Bartlett, Foster, and Telgarsky (and then the PAC-Bayes versions by Neyshabur et al.) are numerically vacuous empirically, and so whether those bounds or these bounds for RNNs explain generalization is up for debate.